# Autophagy regulates fatty acid availability for oxidative phosphorylation through mitochondria-endoplasmic reticulum contact sites

Claudie Bosc[1,2,12], Nicolas Broin[1,2,12], Marjorie Fanjul [1,2], Estelle Saland[1,2], Thomas Farge[1,2], Charly Courdy[1,2], Aurélie Batut[3], Rawand Masoud[4], Clément Larrue[5], Sarah Skuli[1,2,11], Nicolas Espagnolle [6], Jean-Christophe Pagès[6], Alice Carrier[4], Frédéric Bost[7], Justine Bertrand-Michel[3], Jérôme Tamburini[5,8], Christian Récher [1,2,9], Sarah Bertoli [1,2,9], Véronique Mansat-De Mas[1,2,10], Stéphane Manenti[1,2], Jean-Emmanuel Sarry [1,2,12✉] & Carine Joffre [1,2,12✉]

Autophagy has been associated with oncogenesis with one of its emerging key functions being its contribution to the metabolism of tumors. Therefore, deciphering the mechanisms of how autophagy supports tumor cell metabolism is essential. Here, we demonstrate that the inhibition of autophagy induces an accumulation of lipid droplets (LD) due to a decrease in fatty acid β-oxidation, that leads to a reduction of oxidative phosphorylation (OxPHOS) in acute myeloid leukemia (AML), but not in normal cells. Thus, the autophagic process participates in lipid catabolism that supports OxPHOS in AML cells. Interestingly, the inhibition of OxPHOS leads to LD accumulation with the concomitant inhibition of autophagy. Mechanistically, we show that the disruption of mitochondria–endoplasmic reticulum (ER) contact sites (MERCs) phenocopies OxPHOS inhibition. Altogether, our data establish that mitochondria, through the regulation of MERCs, controls autophagy that, in turn finely tunes lipid degradation to fuel OxPHOS supporting proliferation and growth in leukemia.

[1] Cancer Research Center of Toulouse (CRCT), INSERM U1037, CNRS ERL5294, University of Toulouse, Toulouse, France. [2] Equipe labellisée, La Ligue contre le Cancer, Toulouse, France. [3] MetaToul-MetaboHUB, National Infrastructure of Metabolomics and Fluxomics, Toulouse F-31077, France. [4] Aix Marseille Université, CNRS, INSERM, Institut Paoli-Calmettes, Centre de Recherche en Cancérologie de Marseille, Marseille, France. [5] Translational Research Centre in Onco-hematology, Faculty of Medicine, University of Geneva, 1211 Geneva, Switzerland. [6] STROMALab, Université de Toulouse, CNRS ERL5311, EFS, INP-ENVT, Inserm U1031, UPS, Toulouse, France. [7] Inserm U1065, C3M, Team Cellular and Molecular Physiopathology of Obesity and Diabetes, Université Nice Côte d'Azur, Inserm, Nice, France. [8] Université de Paris, Institut Cochin, CNRS UMR8104, INSERM U1016, F-75014 Paris, France. [9] Service d'hématologie, Institut Universitaire du Cancer de Toulouse-Oncopole, Toulouse, France. [10] Laboratoire d'Hématologie, Institut Universitaire du Cancer de Toulouse-Oncopole, Toulouse, France. [11] Present address: Division of Hematology and Oncology, Hospital of The University of Pennsylvania, Philadelphia, PA, USA. [12] These authors contributed equally: Claudie Bosc, Nicolas Broin, Jean-Emmanuel Sarry, Carine Joffre. ✉email: jean-emmanuel.sarry@inserm.fr; carine.joffre@inserm.fr

Autophagy is a dynamic catabolic process in which cytoplasmic components, including proteins and organelles, are sequestered into specific intracellular vesicles, called autophagosomes, before to be delivered to lysosomes for degradation[1]. However, autophagy is not only a sink since degradation products are reused by the cell to sustain metabolism and allow cell survival. Autophagy is a sensor of the metabolic state and allows cells to adapt their demands to poor growth environments, and is considered as a crucial survival mechanism[2]. In physiological conditions, autophagy is for instance required to provide amino acids to the starving neonates after birth[3] or to maintain circulating glucose levels in fasting adult mice[4].

Autophagy has also been critically implicated in tumorigenesis. This catabolic process was first shown to favor cancer progression[5,6], at least in part through its capacity to support the exacerbated metabolism of cancer cells, and therefore to promote their proliferation and survival[7–11]. Autophagy was then proposed to fuel tumor cell metabolism by supplying metabolic substrates, such as glucose and amino acids to maintain either glycolytic capacity[8] or mitochondrial functions depending on cancer types[12–14]. However how does autophagy control the nature and the availability of the substrates, and how is autophagy itself regulated at the molecular level remain largely unknown.

Tumor cells can use diverse oxidizable substrates to fuel their energy metabolism[15]. For instance, fatty acid oxidation (FAO) is a crucial catabolic pathway for cell proliferation in acute myeloid leukemia (AML)[16], and chemotherapy-resistant AML cells exhibit a high OxPHOS status largely dependent on FAO[17,18]. These metabolic changes open new therapeutic avenues and identify mitochondrial metabolism as an important target in AML[19–22]. While the importance of autophagy in AML cell proliferation was recently reported in vitro and in vivo[6,23], the contribution of autophagy to AML metabolism and more specifically to fatty acid metabolism is currently unknown.

Here, we show that autophagy (i.e., lipophagy) maintains energy metabolism by suppling free fatty acids (FFAs) to mitochondria through the degradation of LD in AML cells, but not in normal hematopoietic cells. Furthermore, we demonstrate that mitochondria modulate lipid catabolism through the regulation of the autophagy process. Our study also reveals that the control of autophagy by the mitochondria, and then the regulation of lipid availability require the maintenance of the tethering between the ER and the mitochondria membranes referred as the mitochondria–ER contacts or MERCs[24]. Thus, this study reports a new regulatory loop, in which mitochondria control their own energy and respiratory sources through the regulation of autophagosome formation at MERCs, necessary to AML cell proliferation and survival in vitro and in vivo.

## Results

**Autophagy participates to lipid catabolism to support OxPHOS.** Whether FFAs might represent a critical substrate to fuel OxPHOS in AML cells compared to their normal healthy counterpart cells (hereafter normal hematopoietic cells) is still not known. To address this question, AML blasts (from peripheral blood; Supplementary Table 1) and normal hematopoietic cells (peripheral blood mononuclear cells (PBMC) and CD34+ cells; Supplementary Table 2) were first treated with a low concentration of etomoxir (Etx, 3 μM) to prevent the entry of FFAs into mitochondria by blocking the activity of carnitine palmitoyl transferase 1, and oxygen consumption rate (OCR) was evaluated by Seahorse analysis. We found that the OCR (Fig. 1a, Supplementary Fig. 1a, b) and ATP production linked to respiration (Supplementary Fig. 1d) were significantly decreased up to 30–50% in AML cells from two cell lines (MOLM14 and U937)

and in primary patient samples compared to respective control AML cells. Conversely, basal OCR and ATP-linked OCR in the presence of Etx were only slightly decreased in primary normal hematopoietic cells (Fig. 1a, Supplementary Fig. 1c, d). Therefore, FFAs represent a minor substrate in normal hematopoietic cells, but are an important carbon source that fuels the TCA cycle and oxidative phosphorylation in AML cells.

We next addressed the origin of FFAs in AML. Autophagy can generate FFAs via the degradation of LD[25,26] and has been implicated in tumor lipid homeostasis[27]. As autophagy has also been shown to be required for AML cell proliferation[6], we hypothesized that autophagy also contributes to lipid catabolism to support TCA cycle and mitochondrial energetic metabolism in AML cells. To test this hypothesis, two human AML cell lines were first treated with a well-known pharmacologic inhibitor of autophagosome formation, 3-methyladenine (3-MA), and lipid content was analyzed by flow cytometry using the BODIPY 493/503 probe. Treatment with 3-MA revealed a significant increase in fluorescence intensity as compared to control conditions, indicating an accumulation of lipids after inhibition of autophagy (Supplementary Fig. 1e). Consistent with these results, microscopy studies revealed a cytosolic accumulation of lipid in structures resembling LD in AML cells treated with 3-MA, with no change in lipid content in normal hematopoietic cells (Fig. 1b–d, Supplementary Fig. 1f). This suggests that the involvement of autophagy in lipid metabolism of normal hematopoietic cells is negligible compared to AML cells. To rule out 3-MA off-target effects, we investigated the consequences of short-term and long-term silencing of the key autophagy proteins Beclin1 and ATG12 on lipid accumulation. Similar to 3-MA treatment, the number and area of BODIPY 493/503-positive structures increased upon genetic inhibition of autophagy in AML cells (Fig. 1e–g, Supplementary Fig. 1g). Of note, no change in lipid content of normal hematopoietic cells was significantly observed (Supplementary Fig. 1h, i). This demonstrates that autophagy regulates lipid content in AML cells. Furthermore, specific acid lipase lysosomal inhibitor Lalistat2 also promoted the accumulation of BODIPY 493/503-positive structures in AML cells (Supplementary Fig. 1j, k). To further characterize these intracellular organelles, we quantified content in triglycerides, one of the major storage lipids sequestered in LD. We observed a significant increase in triglyceride level in autophagy-incompetent cells (Fig. 1h). These results combined with the intracellular punctuated BODIPY staining indicate that inhibition of autophagy led to the accumulation of triglycerides in LD in AML cells. Altogether these experiments strongly suggest that autophagy is involved in lipid catabolism in AML cells to generate FFAs necessary for mitochondrial activity, and that this process is not observed in normal hematopoietic cells, where lipids are not a major respiratory source.

Since fatty acid oxidation (FAO) is the biochemical process responsible for FA degradation and indicative of the level of FFAs available to cells, we next analyzed FAO in AML cells. As expected, the inhibition of autophagosome formation by an shRNA directed against ATG12 (Fig. 1i) or by 3-MA (Supplementary Fig. 1l) reduced the rate of FAO. Thus, autophagy of LD (i.e., lipophagy) contributes at least in part to the release, and availability of FFAs from triglyceride hydrolysis in AML cells. We therefore studied whether the FFAs released by autophagy and utilized by mitochondria represent an important respiratory substrate that participates in ATP production by OxPHOS. To do so, we assessed OCR and mitochondrial ATP production in cells with inhibited autophagy. This approach revealed that both basal OCR and mitochondrial ATP production-linked OCR were significantly decreased, when autophagy was inhibited either with 3-MA (Fig. 1j, Supplementary Fig. 1m, n) or with a small

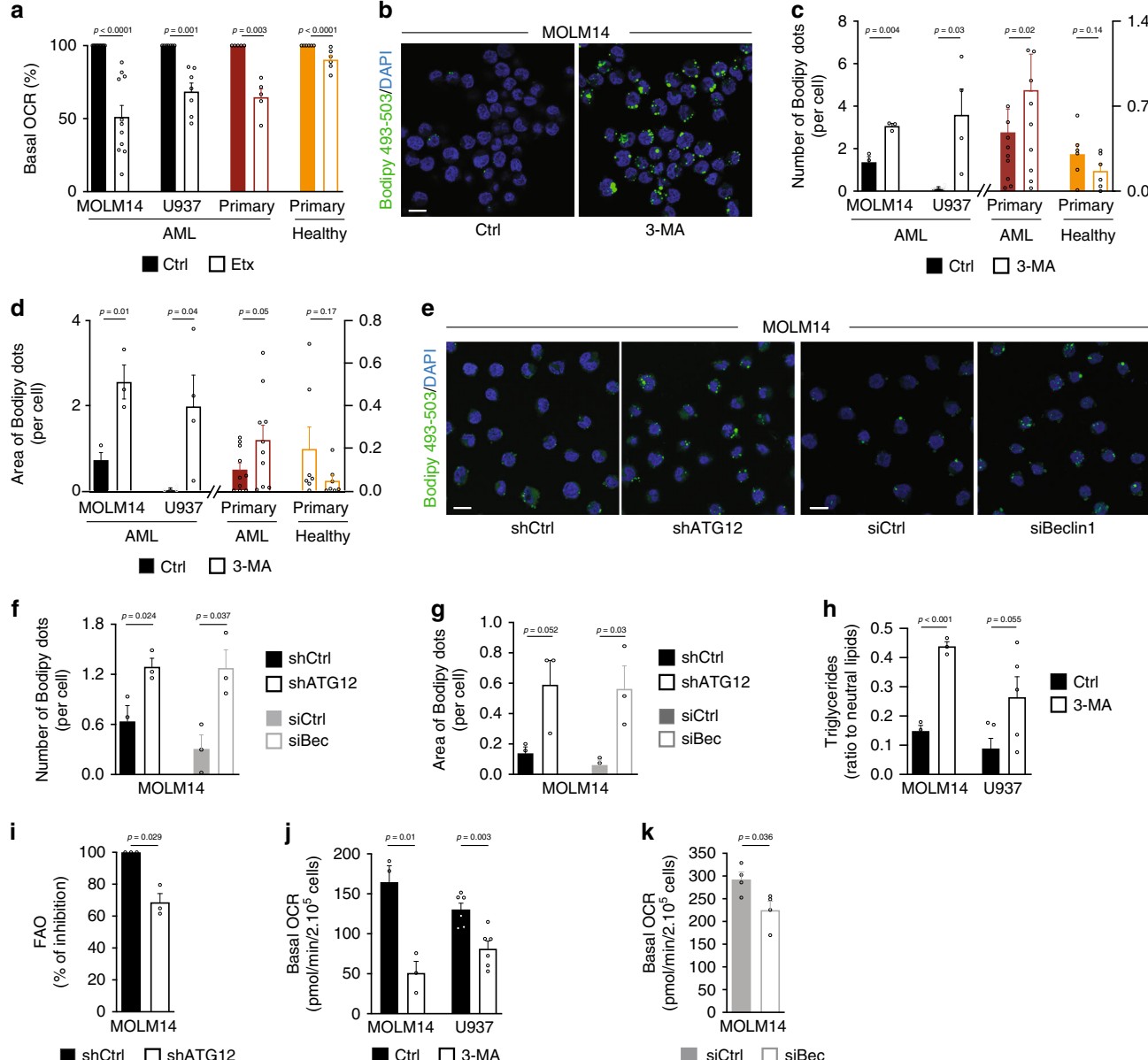

**Fig. 1 Autophagy participates to lipid catabolism to support OxPHOS. a** Seahorse measurement of basal oxygen consumption rate (OCR) in MOLM14 ($n = 11$) and U937 ($n = 7$) AML cell lines, in primary AML patient cells ($n = 4$) and in primary normal hematopoietic cells (PBMC $n = 6$; CD34$^+$ $n = 3$) treated or not with Etx (3 μM, 15 min; one-sample $t$-test). **b** MOLM14 cells were treated with 3-methyladenine (3-MA, 5 mM, 24 h), fixed and stained for Bodipy 493/503 and DAPI. Representative confocal pictures from three independent experiments are shown. Scale bar: 10 μm. **c, d** MOLM14 ($n = 3$) and U937 ($n = 4$) AML cell lines, primary AML patient cells ($n = 10$) and primary normal hematopoietic cells (PBMC $n = 7$) were treated or not with 3-MA (5 mM, 24 h), fixed, and stained for Bodipy 493/503 and DAPI. Histograms show the number (**c**) or the area (**d**) of Bodipy 493/503 dots per cell (one-sample $t$-test). **e, g** MOLM14 cells were either transduced with a shRNA directed against ATG12 or transfected with a siRNA targeting Beclin1. Cells were then stained for Bodipy 493/503 and DAPI. Representative confocal pictures from three independent experiments are shown (**e**). Scale bar: 10 μm. Graphs represent the number (**f**) or the area (**g**) of Bodipy 493/503 dots per cell, ($n = 3$, unpaired $t$-test). **h** MOLM14 ($n = 3$) and U937 ($n = 5$) cells were treated with 3-MA (5 mM, 24 h), and processed for triglycerides content analysis. Graph represents the ratio of triglycerides on total neutral lipids (unpaired $t$-test). **i** MOLM14 transduced with Ctrl or ATG12 shRNAs were examined for their rates of β-oxidation, ($n = 3$, one-sample $t$-test). **j, k** Seahorse measurement of basal OCR in MOLM14 ($n = 3$) and U937 ($n = 6$) cells treated or not with 3-MA for 24 h (**j**), or MOLM14 cells transfected with siRNA control or targeting Beclin1 ($n = 4$) (**k**) (unpaired $t$-test). Data are means ± s.e.m.

interfering RNA (siRNA) targeting Beclin1 in AML cells (Fig. 1k, Supplementary Fig. 1o, p). Conversely, OCR and mitochondrial ATP production were slightly decreased in normal hematopoietic cells (Supplementary Fig. 1q, r). In summary, these results indicate that AML cells in part rely on autophagy for energy metabolism by supplying FFAs to fuel oxidative phosphorylation.

**Inhibition of OxPHOS affects lipid metabolism**. Given that our findings established that FFAs are essential for mitochondrial activity, we next wondered whether OxPHOS could in turn regulate the lipid metabolism. Transcriptomic analysis of AML cells treated with the mitochondrial electron transfer chain (ETC) complex I inhibitor metformin[18] (Supplementary Fig. 2a) showed a downregulation of genes regulating metabolic processes (Fig. 2a,

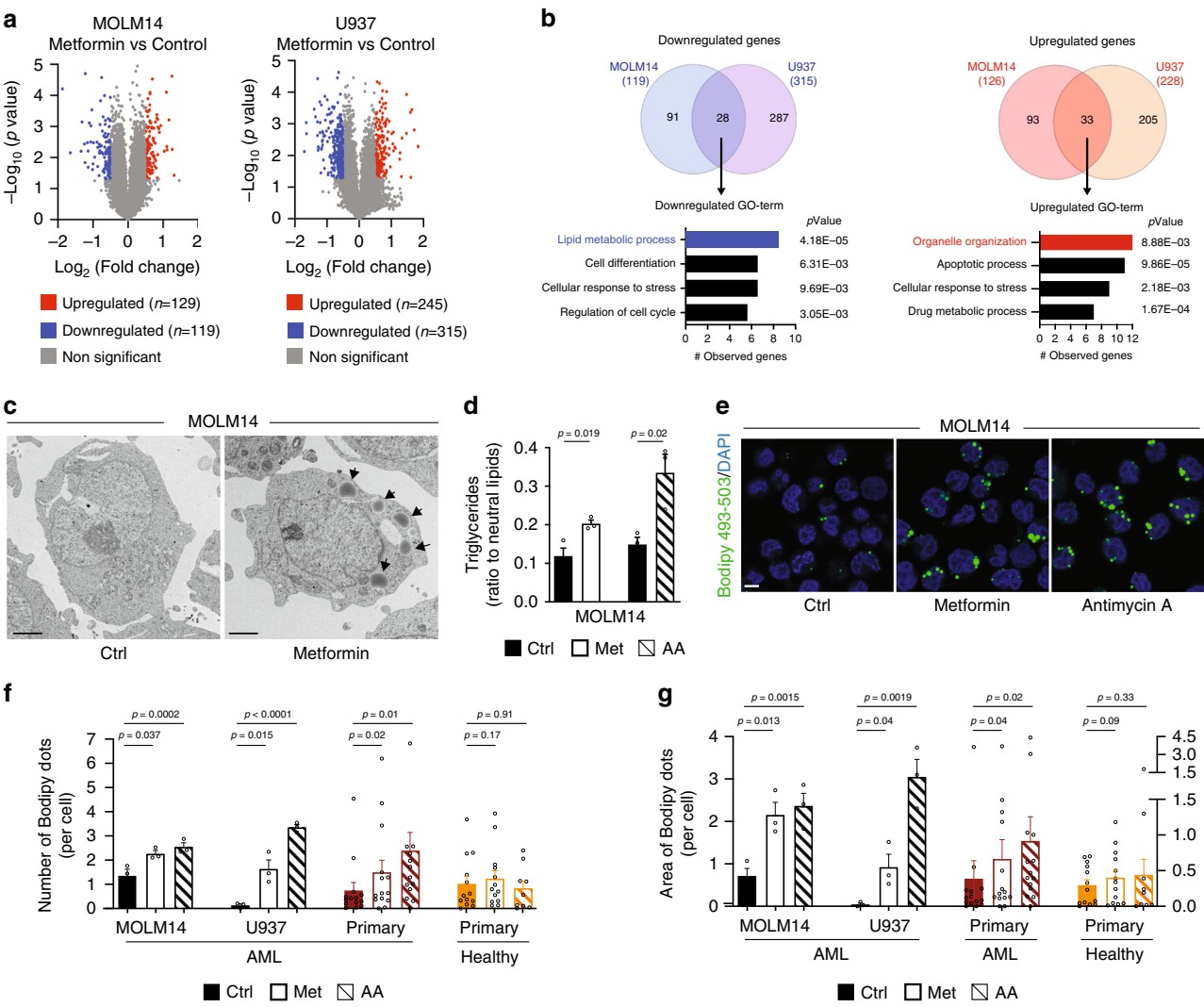

**Fig. 2 Inhibition of OxPHOS affects lipid metabolism. a** Volcano plots displaying fold change versus adjusted p-values of MOLM14 (left) and U937 (right) cells treated with 10 mM of metformin (Met) for 24 h (p-value < 0.05, absolute log2 fold change > 0.5, unpaired t-test). **b** Venn diagram representing overlap between MOLM14 and U937 downregulated (left) and upregulated (right) genes. Enrichment analysis of Gene Ontology (GO) classification for common downregulated (left) and upregulated (right) genes by Genomatics software analysis (Fisher's exact test). **c** MOLM14 cells treated with Met (10 mM) for 24 h were fixed and processed for transmission electron microscopy analysis. Representative electron microscopy pictures from two independent experiments are shown. Arrows indicate lipid droplets. Scale bar: 2 μm. **d** MOLM14 cells were treated with Met or with antimycin A (AA) and processed for triglycerides content analysis, (n = 3, unpaired t-test). **e** MOLM14 cells were treated with Met or with AA, fixed and stained for Bodipy 493/503 and DAPI. Representative confocal pictures from three independent experiments are shown. Scale bar: 10 μm. **f, g** MOLM14 (n = 3) and U937 (n = 3) AML cell lines, primary AML patient cells (n = 14) and primary normal hematopoietic cells (PBMC n = 9; CD34+ n = 4) were treated with Met or with AA for 48 h, fixed, and stained for Bodipy 493/503 and DAPI. Histograms show the number (**f**) or the area (**g**) of Bodipy 493/503 dots per cell (unpaired t-test or paired t-test for patients' samples). Data are means ± s.e.m.

b, Supplementary Tables 3 and 4), and in particular lipid metabolic processes (Fig. 2b, Supplementary Table 5). We then performed a gene set enrichment analysis (GSEA) with two identified fatty acid metabolism gene signatures (Hallmark fatty acid metabolism signature, 158 genes, Broad Institute M5935; fatty acid metabolic process signature[28], 47 genes) and observed an enrichment of these two gene signatures in control-treated cells compared to metformin-treated cells (Supplementary Fig. 2b), indicating that these specific gene sets related to lipid metabolism were downregulated upon metformin treatment. Moreover, high OxPHOS gene signature that we have previously determined[18] was negatively selected in metformin-treated cells (Supplementary Fig. 2c). Therefore, the inhibition of mitochondrial ETC complexes was associated with a decrease in lipid metabolism, supporting a link between mitochondrial activity and lipid

metabolism. An electron microscopy study further revealed that metformin treatment induces the appearance of structures corresponding to LD that were entirely absent in control-treated cells (Fig. 2c, Supplementary Fig. 2d). Accordingly, level of triglycerides was significantly increased in AML cells treated with metformin, as well as with a specific ETC complex III inhibitor antimycin A (AA) compared with control cells (Fig. 2d, Supplementary Fig. 2e). To further investigate the relationship between mitochondrial activity and lipid homeostasis, we performed flow cytometry analysis with BODIPY 493/503 and demonstrated that lipid staining increased over time upon metformin or AA treatment (Supplementary Fig. 2f). We then assessed the number of lipid droplets present after mitochondrial OxPHOS inhibition in two AML cell lines, primary AML patient cells and normal hematopoietic cells. As observed with autophagy inhibition

(Fig. 1b–g, Supplementary Fig. 1f, h, i), mitochondrial respiratory chain inhibition led to an accumulation of LD in AML cells and had no impact on normal hematopoietic cells (Fig. 2e–g, Supplementary Fig. 2g). Of note, similar results were obtained with cells grown in lipid-free serum medium, suggesting that the increase in LD resulted from the accumulation of both extracellular and endogenous lipids (Supplementary Fig. 2h, i).

Next, we examined whether this accumulation was due to an increased lipogenesis or a decrease in degradation. As previously reported[29], metformin strongly reduced de novo lipogenesis (Supplementary Fig. 2j). Importantly, our transcriptomic analysis did not reveal any increase in gene expression, and signatures related to the biogenesis or the trafficking of LD, to the synthesis of triglycerides, or to transcription factors regulating lipid synthesis (Fig. 2b, Supplementary Tables 3–5). In addition, no change in the expression level of proteins implicated in LD formation (i.e., ADRP; Supplementary Fig. 2k), or modification of the subcellular localization of the transcription factor SREBP1/2 were observed upon metformin or AA treatment (Supplementary Fig. 2l). Moreover, both metformin and AA markedly reduced FAO (Supplementary Fig. 2m), suggesting that the inhibition of OxPHOS also prevented the lipid degradation. Collectively, these data indicate that mitochondrial OxPHOS is directly linked to the lipid metabolism in AML cells by controlling the degradation of LD and the availability of respiratory substrates.

**Inhibition of OxPHOS reduces autophagic flux.** Based on above-described results, we next sought to determine whether mitochondrial function might regulate lipid degradation via the control of the autophagy process. To address this possibility, we evaluated if inhibiting the mitochondrial ETC could modulate autophagy. To measure the autophagic flux, the conversion of LC3B-I to LC3B-II that reflects the number of autophagosomes[30] was assessed by immunoblotting in presence or absence of chloroquine (chloro), an inhibitor of lysosomal degradation. These experiments revealed that LC3B-II accumulation was significantly (>50% after 48 h) decreased in metformin- or AA-treated AML cells from cell lines and primary patient specimens (Fig. 3a–f, Supplementary Fig. 3a–c), but not in normal hematopoietic cells (Fig. 3e, f). We confirmed the impact of ETC on autophagy with immunofluorescence studies of endogenous LC3B. The number of autophagosomes per cell was significantly decreased upon metformin treatment compared to controls in all AML cells, without modification in number of autophagosomes detected in healthy cells (Fig. 3g–i, Supplementary Fig. 3d–f). Furthermore, flow cytometry analysis of autophagic vacuoles using Cyto-ID assay fully confirmed this observation in AML cells (Supplementary Fig. 3g). Altogether these approaches consistently support the contention that mitochondrial ETC and OxPHOS activity positively regulate autophagy in AML cells, but not necessarily in normal hematopoietic cells.

**OxPHOS regulates MERCs number and function.** These above-mentioned results demonstrate that the capacity of mitochondria to regulate the lipid availability through autophagy seems unique to tumor cells. Therefore, we next investigated how mitochondria mechanistically control autophagy in AML cells. Electron microscopy analysis of AML cell lines showed that ~20% of the mitochondria were in close proximity to the ER (Fig. 4a, b, Supplementary Fig. 4a, b). These inter-organelle contacts between mitochondria and the ER called MERCs are now recognized as essential regulators of fundamental cellular processes, including autophagy[31], bioenergetics[32], and as key players in oncogenesis[33]. Interestingly, metformin markedly reduced the number of these MERCs (Fig. 4a, b, Supplementary Fig. 4a, b) without affecting

the length (MOLM14: 204 nm ± 22; MOLM14 + Met: 228 nm ± 20; U937: 361 nm ± 36; U937 + Met: 253 nm ± 27), the number or mass of mitochondria (Supplementary Fig. 4c, d). Expectedly, since one of the key functions of MERCs is to regulate calcium homeostasis, both metformin and AA reduced the mitochondrial calcium content. This thus suggests a MERCs-dependent diminished $Ca^{2+}$ transfer from ER to mitochondria (Supplementary Fig. 4e). In addition, MERCs support autophagy by the recruitment of protein–protein complexes implicated in the autophagosome formation[34,35]. To next validate the role of MERCs in autophagosome formation in AML cells, we developed a protocol based on Wieckowski et al. subcellular fractionation procedure[36]. We fractionated extracts of AML cells and characterized the MERCs fraction by the presence of FALC4 (Fig. 4c). Of note, the absence of detection of VDAC1 and IP3R1 in this fraction was likely due to the relative low amount of proteins obtained at the end of the procedure. This analysis allowed us to determine the presence of two proteins implicated in autophagosome biogenesis in these enriched MERCs fractions by Western blotting, Vps34 and ATG16L (Fig. 4c). We therefore hypothesized that the inhibition of autophagy observed upon ETC inhibition in AML cells was due to a reduction of MERCs. To explore this, we first used the proximity ligation assay (PLA) to investigate mitochondria–ER contacts as described by Tubbs et al.[37]. We used one antibody targeting the inositol trisphosphate receptor (IP3R1) located on the ER, and another one targeting the voltage-dependent anion channel (VDAC1) present in the outer membrane of mitochondria, both proteins (IP3R1 and VDAC1) are located at MERCs. Interactions between the two organelles were visualized by red dots as observed in control cells (Fig. 4d). Consistent with our electron microscopy data, we confirm that metformin or AA significantly decreased the number of MERCs compared to vehicle-treated cells (Fig. 4e). Interestingly, this loss of contact sites rapidly occurred, as early as 6 h after mitochondrial inhibition (Fig. 4e), prior to detection of autophagy inhibition (Fig. 3a–d). Importantly, this was not due to the decrease in VDAC1 and IP3R1 expression upon metformin and AA treatment (Supplementary Fig. 4f).

**MERCs regulate lipophagy to sustain OxPHOS.** These findings suggest that mitochondrial function appears to regulate MERCs formation and are consistent with the notion that the inhibition of autophagy and the subsequent accumulation of LD observed upon ETC inhibition was due to the loss of MERCs. To validate this possibility and to prevent MERCs formation, we generated AML cells expressing an shRNA directed against VDAC1 or cells depleted for mitofusin2 (Mtfn2) protein by siRNA, a critical player in the formation of MERCs[38]. Electron microscopy studies performed on cells depleted for Mtfn2 or VDAC1 showed that the percentage of mitochondria in close contact with the ER was decreased compared to control cells (Fig. 5a, b, Supplementary Fig. 5a). Of note, the number of mitochondria per cell and the mitochondrial mass between cells depleted or not for Mtfn2 or VDAC1 were roughly the same (Supplementary Fig. 5b, c). Moreover, Mtfn2-depleted cells displayed a significant decrease in PLA dots compared to control cells (Fig. 5c). These results therefore indicate that a loss of Mtfn2 in AML cells led to a reduction of MERCs. We next investigated whether these contact sites modulate autophagy and subsequently lipid metabolism in AML cells. In accordance with the literature[34], the decrease in MERCs resulting from Mtfn2 or VDAC1 depletion decreased the number of autophagosomes (Fig. 5d, e) and the autophagic flux (Supplementary Fig. 5d, e) in AML cells. In addition, the silencing of an ER protein, IP3R1, rather than a mitochondrial protein, reduced also the autophagic flux (Supplementary Fig. 5f). As

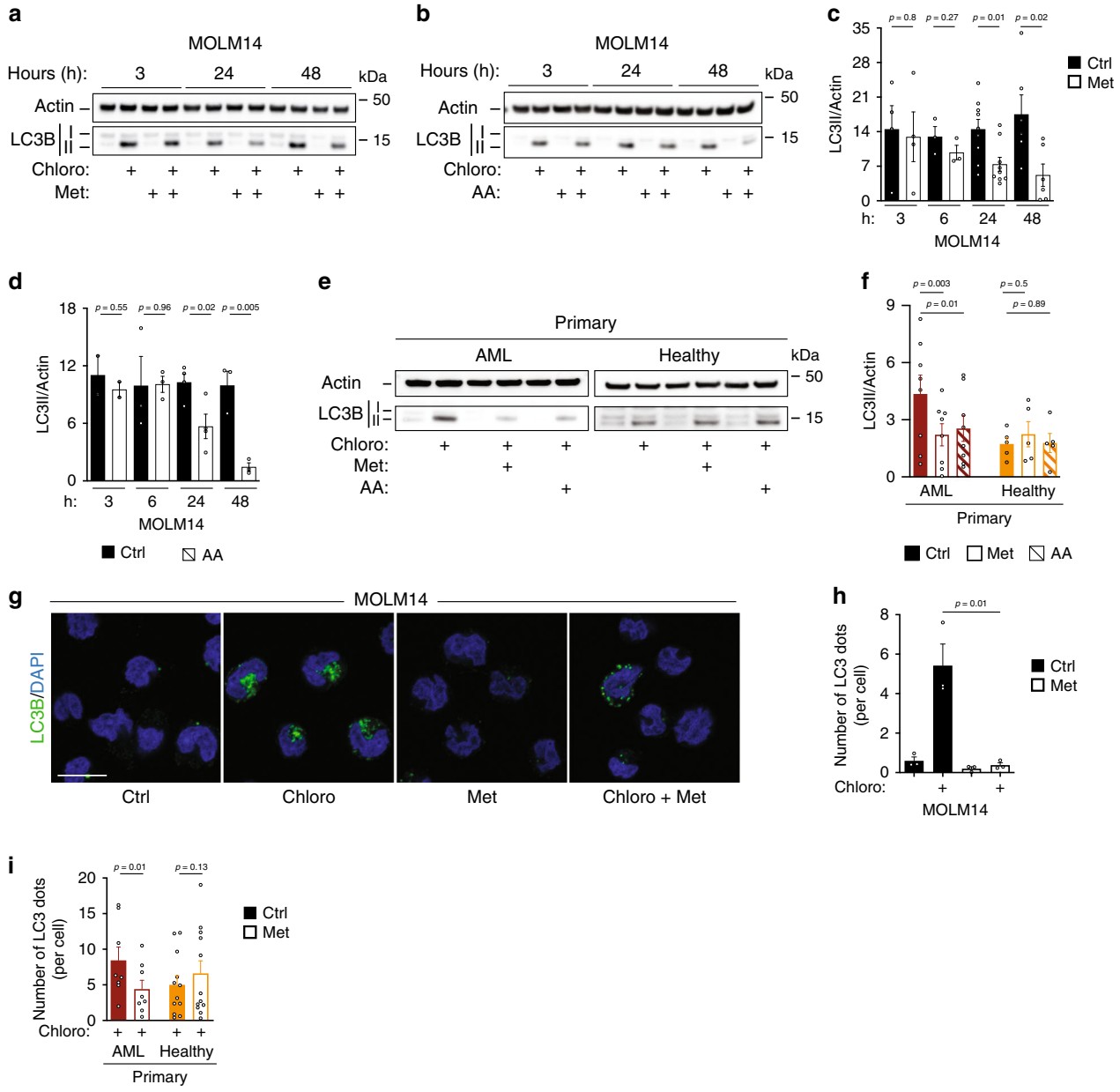

**Fig. 3 Inhibition of OxPHOS reduces autophagic flux. a**, **b** Western blots of LC3B and actin from at least three independent experiments of MOLM14 cells treated with metformin (Met) (**a**) or with antimycin A (AA) (**b**) ± chloroquine (chloro) for the indicated times are shown. **c**, **d** LC3B-II/actin ratios identified by densitometries from Western blots shown in **a**, **b** in MOLM14 cells treated with Met (**c**) or with AA (**d**) in presence of chloro. Data are means ± s.e.m, (at least $n = 3$, unpaired $t$-test). **e** Western blots of LC3B and actin from primary AML ($n = 8$) or normal ($n = 5$) cells treated with Met or with AA ± chloro. **f** Primary AML patient cells ($n = 8$ with Met, $n = 8$ with AA) and primary normal hematopoietic cells (PBMC) were treated with Met ($n = 5$) or AA ($n = 5$) for 48 h in presence of chloro followed by immunoblotting for LC3B and actin. Histograms represent the LC3B/actin ratios obtained by densitometric analysis of Western blots (paired $t$-test). **g**, **h** Representative confocal pictures from three independent experiments of MOLM14 cells treated with Met for 48 h ± chloro, fixed and stained for LC3B and DAPI. Scale bar: 10 μm. **g** Histograms represent the number of LC3B puncta per cell (**h**), ($n = 3$, unpaired $t$-test). **i** Primary AML patient cells ($n = 8$) and primary normal hematopoietic cells (PBMC $n = 8$, CD34$^+$ $n = 4$) were treated or not with Met for 48 h ± chloro, fixed and stained for LC3B and DAPI. Histograms represent the number of LC3B puncta per cell (paired $t$-test). Data are means ± s.e.m.

expected from our hypothesis, fluorescent microscopy analysis showed that both Mtfn2 or VDAC1 depletion led to an accumulation of LD compared to control cells (Fig. 5f, g). Accordingly, electron microscopy analysis showed that Mtfn2- or VDAC1-silenced AML cells accumulated LD (Fig. 5a, Supplementary Fig. 5a). Moreover, since the expression of FASN, ACLY, and ADRP was not modified and no change in the subcellular localization of the transcription factor SREBP1/2 was observed upon Mtfn2 depletion (Supplementary Fig. 5g, h), LD biogenesis is likely not affected with MERCs inhibition. All of these results

phenocopied the effects observed upon metformin treatment (Fig. 2c, Supplementary Fig. 2k, l). Collectively, these data indicate that autophagy inhibition observed upon mitochondrial respiration inhibition was due to a reduction in the number of MERCs and is consistent with mitochondria regulating autophagy through mitochondria–ER contact sites in AML cells. Similar to autophagy inhibition (Fig. 1j, k, Supplementary Fig. 1m–p), the decrease of MERCs upon VDAC1 or Mtfn2 depletion resulted in a decreased OCR (Fig. 5h, Supplementary Fig. 5i, j) and mitochondrial ATP production-linked OCR

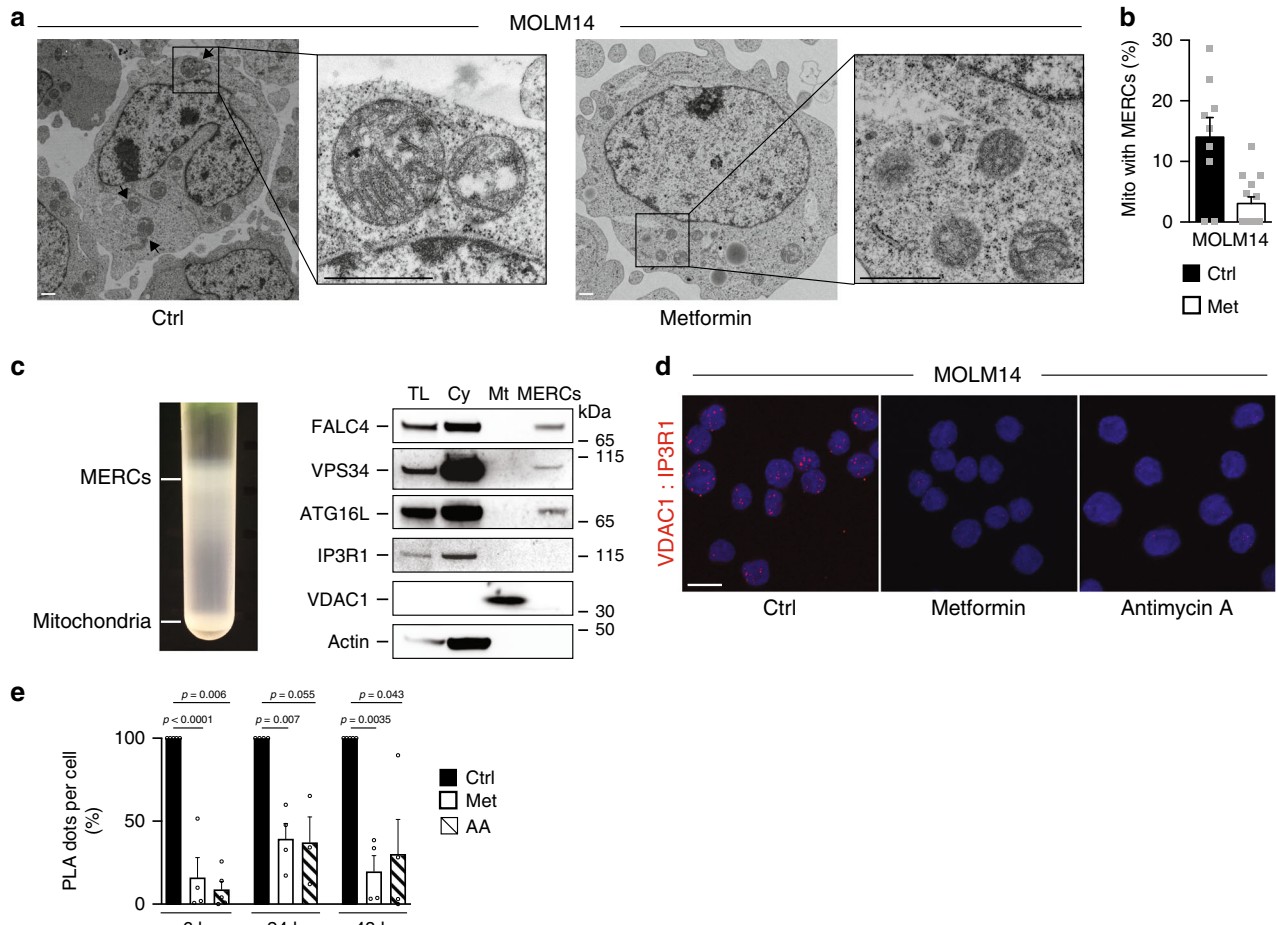

**Fig. 4 OxPHOS regulates MERCs number and function. a** MOLM14 cells treated with metformin (Met, 10 mM) for 24 h were fixed and processed for electron microscopy analysis. Representative electron microscopy pictures from two independent experiments are shown. Arrows indicate MERCs. Scale bar: 1 μm. **b** Histograms represent the % of mitochondria that are in contact with endoplasmic reticulum per cell treated or not with Met from pictures displayed in **a**. Data are means ± s.e.m with each dot corresponding to one cell. **c** Protein components of subcellular fractions (left panel) from two independent experiments prepared from MOLM14 cells revealed by immunoblot analysis (right panel). TL: total lysate, Cy: cytosol, Mt: pure mitochondrial fraction, MERCs: mitochondria–ER contact site fraction. **d** Representative orthogonal confocal projections of Z sections of PLA (red signal) between VDAC1 and IP3R1 from MOLM14 cells treated or not with Met or antimycin A (AA) for 48 h (at least $n = 3$). Scale bar: 10 μm **e** Quantitative analysis of PLA signal between VDAC1 and IP3R1 from MOLM14 cells treated or not with Met or AA for 6 h, 24 h or 48 h ($n = 4$, one-sample $t$-test). Data are means ± s.e.m.

compared to control cells (Supplementary Fig. 5k). Furthermore, when autophagy was inhibited (siBeclin1) or MERCs were disrupted (siMtfn2), mitochondria of AML cells pulsed with a fluorescent fatty acid lipid (RC$_{12}$) exhibited less overlap with this lipid, due to the accumulation of RC$_{12}$ into the cytoplasm (Fig. 5i, j). These results indicate that lipids upon autophagy inhibition or MERCs disruption were not utilized and oxidized by mitochondria. Furthermore, the addition of exogenous fatty acids upon autophagy inhibition or MERCs disruption restored mitochondrial respiration (Fig. 5k). Altogether, these results suggest that MERCs through autophagy regulation control lipid catabolism that supports mitochondrial OxPHOS in AML cells. Since ETC inhibitors reduced MERCs number and functions (Fig. 4, Supplementary Fig. 4), we confirmed that autophagy inhibition and subsequent accumulation of LD in metformin-treated cells are due to the loss of MERCs by using an organelle linker, as performed by Csordas G. et al.[39]. Expression of this organelle linker in MOLM14 cells (OMM-ER) reactivated calcium flux from ER into mitochondria and therefore restored MERCs upon metformin treatment (Supplementary Fig. 5l). This prevented autophagy inhibition and lipid accumulation induced by metformin treatment (Fig. 5l–o). Altogether, these data highlight the role of

MERCs formation regulated by mitochondria activity in autophagosome formation to promote lipid degradation and mitochondrial OxPHOS in AML cells.

**MERCs support the dialog between autophagy and OxPHOS.** Finally, to test the functional relevance of the role of autophagy-dependent MERCs formation, we monitored the proliferation of VDAC1-depleted cells in vitro and in vivo, or we daily treated NOD-SCID-gamma (NSG) immunodeficient mice engrafted with AML MOLM14 cells, with an oral potent ETC complex I inhibitor, the IACS-010759[40]. We first validated that this ETC complex I inhibitor phenocopied in vitro the results observed with other ETC inhibitors metformin or AA on lipid metabolism and autophagy. As expected, IACS-010759-treated cells accumulated LD (Supplementary Fig. 6a, b) and displayed a reduction of the autophagic flux (Supplementary Fig. 6c–f). These experiments showed that VDAC1 depletion or IACS-010759 treatment significantly affected cell proliferation in vitro (Supplementary Fig. 6g, h) and tumor growth in vivo (Fig. 6a, c). Mice engrafted with shVDAC1 AML cells or treated with IACS-010759 displayed a marked reduction of the total tumor cell burden and a significant increase in overall survival compared to mice injected

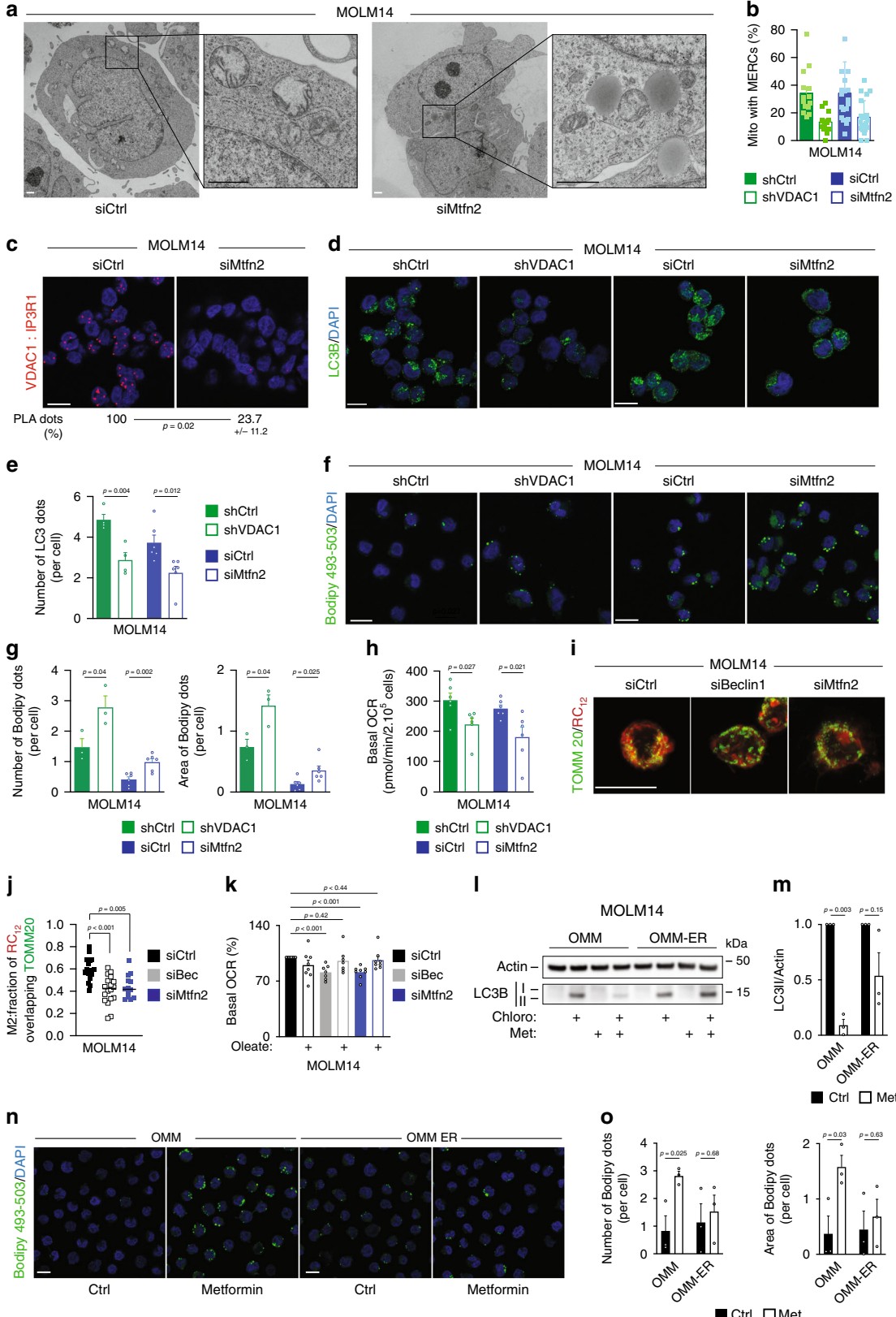

with control cells (Fig. 6b, d). Furthermore, we observed an increase of lipids (Fig. 6e–g), a decrease in autophagy (Fig. 6h) along with a reduction of MERCs (Fig. 6i) in human AML cells from mice treated with IACS-010759, confirming that the observed impact of IACS-010759 in vivo is due to MERCs disruption with subsequent autophagy inhibition and lipid

accumulation. Thus, proliferation and tumorigenicity of AML cells are linked to their ability to produce FFAs via autophagy to ensure mitochondria function that controls in turn autophagy through MERCs formation (Fig. 6j). In conclusion, our data reveal the existence of a finely tuned interplay between mitochondria and autophagy occurring in tumor cells to supply FFAs

**Fig. 5 MERCs regulate lipophagy to sustain OxPHOS. a** MOLM14 cells transfected with siRNA control (Ctrl) or mitofusin2 (Mtfn2) were processed for electron microscopy analysis. Electron microscopy pictures from one experiment are shown. Scale bar: 1 μm. **b** Histograms represent the % of mitochondria in contact with ER. Each dot corresponding to one cell. **c** Images of proximity ligation assay between VDAC1 and IP3R1 from MOLM14 transfected with Ctrl or Mtfn2 siRNAs ($n = 3$). Scale bar: 10 μm. Numbers represent the % of dots in cells transfected with siRNA Mtfn2 compared with cells transfected with siRNA Ctrl. **d, e** MOLM14 cells were transduced with Ctrl or VDAC1 shRNAs ($n = 4$) or transfected with Ctrl or Mtfn2 siRNAs ($n = 6$), and stained for LC3B and DAPI. Confocal sections (at least four independent experiments) are shown. Scale bar: 10 μm (**d**). Histograms represent the number of LC3B puncta per cell (**e**) (unpaired t-test). **f, g** MOLM14 cells transduced with Ctrl or VDAC1 shRNAs ($n = 3$) or transfected with Ctrl or Mtfn2 siRNAs ($n = 6$) were stained for Bodipy and DAPI. Scale bar: 10 μm, (**f**). Histograms show the number or the area of Bodipy dots per cell (**g**) (unpaired t-test). **h** Measurement of basal oxygen consumption rate (OCR) in MOLM14 cells transduced with Ctrl or VDAC1 shRNAs ($n = 6$) or transfected with Ctrl or Mtfn2 siRNAs ($n = 6$, unpaired t-test). **i, j** MOLM14 cells transfected with Ctrl or Beclin1 (Bec) or Mtfn2 siRNAs and incubated with $RC_{12}$, were stained for TOMM20 and confocal Z-stacks were acquired (three independent experiments). Scale bar: 10 μm (**i**). Fraction of $RC_{12}$ overlapping TOMM20 staining (**j**). siCtrl ($n = 17$), siBec ($n = 18$), and siMtfn2 ($n = 13$, unpaired t-test). **k** Measurement of OCR in MOLM14 cells transfected with Ctrl ($n = 8$), Bec ($n = 7$), or Mtfn2 siRNAs ($n = 8$) ± Oleate-BSA. **l–o** MOLM14 cells were transduced with the mitochondria–ER organelle linker (OMM-ER) and its control (OMM), treated ± metformin, and subjected to Western blot analysis for actin and LC3B (**l, m**) or stained for Bodipy and DAPI (**n, o**) (three independent experiments). Scale bar: 10 μm. Histograms represent the number of LC3B puncta (**m**) or the area of Bodipy dots per cell (**o**) unpaired t-test. Data are means ± s.e.m.

to support OxPHOS, and the role of lipids as an essential substrate for AML cell proliferation both in vitro and in vivo (Fig. 6j).

## Discussion

Autophagy is frequently activated in cancer and largely associated with metabolic reprogramming and oncogenesis. Our findings show that autophagy occurring at a specific subcellular location, namely the interconnection between mitochondria and the ER, supplies FFAs to maintain mitochondrial energy metabolism that enables cell proliferation in vitro and in vivo. Furthermore, we found that mitochondrial respiratory chain activity supports autophagy through the regulation of MERCs formation. Altogether these data support a model depicting a bidirectional relationship between autophagy and mitochondria metabolism, in which mitochondria regulates its supply of FFAs by regulating autophagosome formation at MERCs. This relationship allows for the physical and functional integration of LD degradation with the supply of FFAs to fuel the TCA cycle in adjacent mitochondria, and supports the notion that lipid transfer proteins could be located at MERCs (Fig. 6j).

Autophagy could degrade LD[25] and release FFAs that are then oxidized via mitochondrial FAO. ETC inhibitors including metformin are known to regulate autophagy. However, while metformin is mainly described as an autophagy inducer through its agonist action on AMPK[41,42], we report here alongside with two studies[43,44], the unexpected finding that OxPHOS inhibitors such as metformin can interfere with and inhibit autophagosome formation. Moreover, as opposed to what it was shown in adipocytes and hepatocytes[45,46], we showed that metformin inhibits lipid degradation and FAO in AML cells. This occurred at less extend in normal hematopoietic cells.

Cumulatively, our results indicate that the inhibition of autophagy and the subsequent accumulation of LD observed upon ETC inhibition are a consequence of MERCs disruption. Upon ETC inhibition, MERCs alteration is an early event (6 h) before the reduction of autophagy (24 h) and the accumulation of LD at later stage (48 h). In addition, restoring MERCs formation and calcium uptake by the mitochondria using the mAKAP1-mRFP-yUBC6 (i.e., OMM-ER) organelle linker[39] prevented metformin-induced autophagy inhibition and LD accumulation. MERCs have been previously reported as an important site of autophagosome formation in cancer cells[34], fibroblasts[35], and in epithelial cells[47]. However, the role of MERCs in autophagosome formation was not yet described in AML cells. Then, the constitutive presence of key molecules involved in the early events of autophagic flux (Vps34 and ATG16L, two proteins located on the isolation membrane during the first step of autophagosomes formation) in purified MERCS confirms their crucial roles in autophagosome formation and in AML autophagy.

Cancer cells usually exhibit an exacerbated metabolism compared to their normal counterpart. Even if this metabolic reprogramming, known as the Warburg effect, originally consisted to use the glycolysis to produce energy, a vast majority of cancer cells rely rather on mitochondrial oxidative metabolism[48]. Several solid and blood cancers have dependency on OxPHOS and/or FAO pathways either in steady state[28,49,50] and in metastasis[51,52], but also upon treatments[17,18]. This suggest that understanding better the contribution of respiratory sources, such as FFAs to mitochondrial OxPHOS[44] will be crucial to deeper fight against cancer, drug resistance, and relapse. Therefore, our study identified the interplay between mitochondrial metabolism and autophagy as a critical regulator in AML cell proliferation, and could represent a potential therapeutic target for AML patients. This should be further explored in other cancers that rely on mitochondrial metabolism[53,54].

## Methods

**AML cell lines.** The human myeloid leukemia cell lines, MOLM14 and U937 were purchased from the Leibniz Institute DSMZ-German Collection of Microorganisms and Cell Cultures (Leibniz, Germany). MOLM14 shATG12 were previously generated in our laboratory. MOLM14 CRISPR CTL or AMPK-KO were a gift from Prof. Jérome Tamburini (University of Geneva, Switzerland). Cells were grown in minimum essential medium-α medium with Glutamax (Gibco, Life Technologies) supplemented with 10% fetal calf serum (Sigma).

**Primary AML cells and normal hematopoietic cells.** Primary AML patient cells from peripheral blood have been collected during routine diagnostic procedures at the Toulouse University Hospital, after informed consent and stored at the HIMIP collection (BB-0033-00060). According to the French law, HIMIP collection has been declared to the Ministry of Higher Education and Research (DC 2008-307 collection 1) and obtained a transfer agreement (AC 2008-129) after approbation by the "Comité de Protection des Personnes Sud-Ouest et Outremer II" (ethical committee). PBMC were obtained from blood samples of healthy donors (Etablissement Français du Sang, EFS, Toulouse, France). CD34 positive cells were obtained after sorting from umbilical cord blood samples (Etablissement Français du Sang, EFS, Besançon, France). Normal hematopoietic cells (PBMC and CD34⁺) were obtained from EFS that is a governmental agency collecting and delivering blood products, all procedures in use at EFS are defined by the Law. For samples aimed at research use a personal "informed consent" form is signed at the time of collection. This form was validated by the "Agence de la Biomedecine" the body that, in France, rules all type of samples of human origin, on ethical and practical aspects, for research or clinical applications. Clinical and biological annotations of the samples have been declared to the CNIL (Comité National Informatique et Libertés, i.e., Data processing and Liberties National Committee). Briefly, mononuclear cells were separated by Ficoll–Hypaque density gradient centrifugation and incubated in RBC lysis buffer (ammonium chloride solution) to remove red blood cells. CD34⁺ cells isolation was performed using manufacturer's instructions (EasySep™, STEMCELL). Primary AML, PBMC, and CD34⁺ samples were

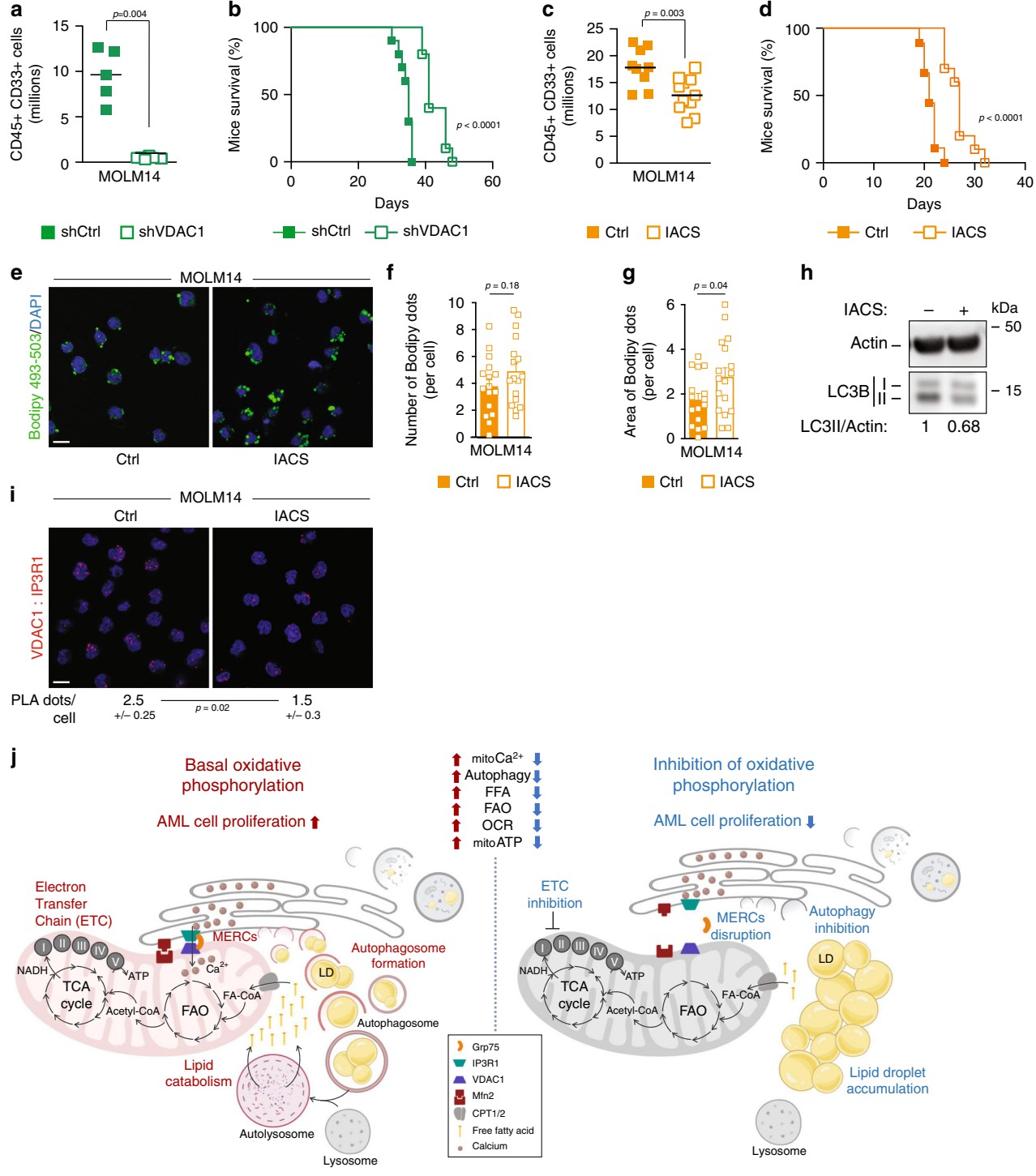

maintained in Iscove's Modified Dulbecco's Medium supplemented with 20% fetal calf serum.

**Antibodies and reagents**. The following antibodies from Cell Signaling Technology were used: rabbit antibodies against LC3B (#2775), ATG12 (#4180), FASN (#3180), Beclin1 (#3738), and mouse antibodies against HSP90 (#4874). Mouse antibodies from Santa Cruz Biotechnology against IP3R1 (sc-271197) and E2F1 (KH95), from Millipore against actin, rabbit antibody from Abcam against VDAC1 (ab15865), from PROGEN against ADRP (610102), from R&D System against FABP4 (AF3150), from Atlas Antibodies against ACLY (HPA022434), from Novus Biologicals against SREPB1 (NB600-582), from Invitrogen against SREBP2 (PA1-338), and from GeneTex against TOMM20 (GTX133756) were also used. Secondary antibodies labeled with horseradish peroxidase were purchased from Promega and those coupled with fluorophores were purchased from Invitrogen. Rabbit anti-LC3B from MBL (#PM036) was used for immunofluorescence analysis. For flow cytometry studies, the antibodies anti-hCD45-APCH7 (BD Biosciences; 641417), CD33-PE (BD Biosciences; 555450), Annexin-V-V500 (BD Biosciences; 561501), and BD Via-Probe™ (BD Pharmingen) were used. Bafilomycin (20 nM) and 3-MA (5 mM) were purchased from InvivoGen. Chloroquine (20 μM), metformin (10 mM), antimycin (10 μM), and lalistat2 (20 μM) were obtained from Sigma Aldrich. Bodipy™ 493/503 probe was purchased from Thermo-FisherScientific. Etx (3 μM, HY-50202A) was purchased from Medchem. EBSS (24010-43, Gibco) was used to induce autophagy.

**Subcellular fractionation**. Buffer A: 20 mM Hepes pH7.4, 10 mM NaCl, 1.5 mM MgCl2, 0.2 mM EDTA, 20% glycerol, 1 mM DTT, 0.1% NP-40, and protease inhibitors.

Buffer B: 20 mM Hepes pH7.4, 500 mM NaCl, 1.5 mM MgCl2, 0.2 mM EDTA, 20% glycerol, 1 mM DTT, 0.1% NP-40, and protease inhibitors.

**Fig. 6 MERCs support the dialog between autophagy and OxPHOS. a, b** NSG mice ($n = 31$) were engrafted with MOLM14 cells expressing the Ctrl or VDAC1 shRNAs by intravenous injection. Seventeen days post graft, five mice shCtrl and six mice shVDAC1 were killed, and the number of human cells (hCD45+ and hCD33+) in the bone marrow and spleen was analyzed by flow cytometry. Graphs represent the number of human positive cells for hCD45 and hCD33 within the murine bone marrow and spleen from one experiment (**a**) (unpaired $t$-test). The remaining mice per group were used for overall survival analysis (**b**). Graph represents the Kaplan–Meier survival curves (Log-rank test). **c, d** NSG mice ($n = 39$) were engrafted with MOLM14 cells and daily treated with vehicle or IACS-010759 by gavage. Seventeen days post graft, nine mice vehicle group and 11 mice IACS-010759-treated group were killed, and the number of human cells (hCD45+ and hCD33+) in the bone marrow and spleen was analyzed by flow cytometry. Graphs represent the number of human positive cells for hCD45 and hCD33 within the murine bone marrow and spleen (**c**) (unpaired $t$-test). The remaining mice were used for overall survival analysis (**d**). Graph represents the Kaplan–Meier survival curves (Log-rank test). **e–g** Viable bone marrow AML blasts from mice treated with IACS-010759 were stained for Bodipy 493/503 and DAPI. Scale bar: 10 μm (**e**). Histograms show the number or the area of Bodipy 493/503 dots per cell (**f, g**) (unpaired $t$-test). **h** Viable bone marrow AML blasts from mice treated with IACS-010759 (one experiment) were subjected to Western blot analysis for actin and LC3B. Numbers indicate ratios obtained by densitometric analysis. **i** Purified viable bone marrow AML blasts from mice treated with IACS-010759 (one experiment) were subjected to PLA assay. Numbers represent the number of PLA dots per cell (minimum 130 cells, unpaired $t$-test). Scale bar: 10 μm (**j**). Schematic diagram depicting the interplay between the autophagy process and the mitochondria in AML cells. Autophagy appears as a major regulator of mitochondria activity with the mitochondria controlling its supply of FFAs by regulating the number of autophagosomes via the formation MERCs. Data are means ± s.e.m.

Cells were washed with ice-cold phosphate-buffered saline (PBS) pH 7.4. After centrifugation, cell pellets were resuspended in 250 μL of ice-cold buffer A and kept for 10 min in ice. A total of 20 μL of the lysate were removed as "total lysate" and nuclei were pelleted by centrifugation at $500 \times g$ for 5 min at 4 °C.

The supernatant was removed as the "cytosolic fraction" and the pellet of nuclei was gently washed with 350 μL of buffer A and centrifuged at $500 \times g$ for 5 min. The supernatants were discarded. The nuclear pellets were resuspended in 100 μL of hypotonic buffer B and allowed to swell on ice for 30 min. The extract was separated by centrifugation at $21,000 \times g$ for 15 min at 4 °C. The supernatant was collected and designated as "nuclear fraction". All fractions were resuspended in Laemmli sample buffer and "nuclear fractions" and "total lysate" that contained DNA were sonicated.

**Isolation of mitochondria-associated membranes**. MERCs fractions were isolated according to the Nature Protocols from Wieckowski et al.[36]. The cell number and the cell lysis method were adapted for AML cells. Briefly, 2.5 billion of MOLM14 cells were washed with PBS (with Ca2+ and Mg2+) and centrifuged at $600 \times g$ 5 min at 4 °C, twice. Cells were resuspended at 200 millions of cells per mL with the buffer 1 (225 mM mannitol, 75 mM sucrose, 0.1 mM EGTA, and 30 mM Tris HCl pH 7.4) and disrupted using a nitrogen cavitation chamber (PARR Instrument, 7 min at 35 psi). Nuclei and unbroken cells were pelleted by centrifugation at $600 \times g$ for 5 min at 4 °C. After two centrifugations, the pellet was discarded. The supernatant was collected and centrifuged at $7000 \times g$ for 10 min at 4 °C to separate crude mitochondria (pellet) from microsome and ER fractions (supernatant). The crude mitochondrial fraction was suspended in 1 mL of buffer 2 (225 mM mannitol, 75 mM sucrose, and 30 mM Tris HCl pH 7.4). Mitochondrial suspension was centrifuged at $7000 \times g$ for 10 min at 4 °C, and the supernatant was discarded. Mitochondrial pellet was suspended into 1 mL of buffer 2 and centrifuged at $10,000 \times g$ for 10 min at 4 °C. The crude mitochondrial pellet was suspended into 2 mL of MRB buffer (250 mM mannitol, 5 mM HEPES, and 0.5 mM EGTA pH 7.4), layered on top of 8 mL Percoll medium (225 mM mannitol, 25 mM HEPES, pH 7.4, 1 mM EGTA, and 30% Percoll (v/v)), and centrifuged at $95,000 \times g$ for 30 min at 4 °C. The MERCs fraction was collected from Percoll gradient, was washed to remove the Percoll by centrifugation at $6300 \times g$ for 10 min followed by further centrifugation of the supernatant at $100,000 \times g$. Likewise, the pure mitochondria fraction was collected from the bottom of Percoll gradient, washed twice with MRB and centrifuge at $6300 \times g$ for 10 min at 4 °C to obtain a pellet. All the fractions were flash frozen and kept at −20 °C until use.

**Western blot analysis**. Proteins were separated using 4–12% gradient polyacrylamide SDS–PAGE gels (Life Technologies) and electrotransferred to 0.2 μm nitrocellulose membranes (GE Healthcare). After blocking in Tris-buffered saline with 0.1% Tween and 5% bovine serum albumin, membranes were blotted overnight at 4 °C with the appropriate primary antibodies. Primary antibodies were detected using the appropriate horseradish peroxidase-conjugated secondary antibodies. Immunoreactive bands were visualized by enhanced chemiluminescence (PI32209; Thermo Fisher Scientific) with a Syngene camera. Densitometric analyses of immunoblots were performed using the GeneTools software. All full scans of uncropped blots are available in the Supplementary file (Supplementary Fig. 8).

**LC3 flux assay**. LC3B-II/actin ratios identified by densitometric analysis from Western blots were subtracted between ±chloro to get the net LC3 flux between control and treated conditions[30].

**Immunofluorescence analysis**. For LC3B staining, cells were seeded onto glass slides (Fisher Scientific) coated with 0.01% poly-L-lysine (Sigma), then fixed in 4% formaldehyde for 8 min. After PBS washes, cells were incubated in 0.01% saponin containing 3% BSA for 30 min and then incubated with anti-LC3B antibodies (MBL, 1/700) for 45 min. Cells were then washed before incubation with an anti-rabbit Alexa-488 secondary antibody (Invitrogen) for 30 min, followed by PBS and distilled $H_2O$ washes and mounting in ProLong™ Gold antifade medium with DAPI (4′6-diamidino-2-phenylindole, Invitrogen). Images were acquired using a confocal Zeiss LSM 780. For quantification, fields were chosen arbitrarily based on DAPI staining, and the number of LC3B dots per cell of at least 100 cells per independent experiment was determined with Image J software.

**Lipid chase**. For lipid chase experiments, AML cells were labeled with 1 mM of the fluorescent fatty acid BODIPY 558/568 $C_{12}$ ($RC_{12}$Thermo Fisher Scientific) overnight in complete culture medium. After two washes, AML cells were seeded on coverslips, fixed, and stained for mitochondrial network with TOMM20 (Genetex, 133756) like for LC3B staining. For imaging of the mitochondrial network and $RC_{12}$ distribution, 6–8 μm Z-stacks at 0.18 μm were used. Fluorescence signals were analyzed using high resolution fluorescence microscopy. Images were taken with a Zeiss LSM 880 FAST Airyscan using a 63× Plan-Apochromat objective with 1.4 aperture under immersion oil. The Manders 2 (M2) coefficient, or fraction of $RC_{12}$ signal overlapping mitochondrial network, was determined from Z-stack projections of $RC_{12}$ and mitochondrial network using the Fiji JACoP plugin.

**Duolink PLA**. Duolink II PLA (Sigma) enables the detection of protein interactions (<40 nm) as an individual fluorescent dot by microscopy. Cells were fixed 10 min with 4% paraformaldehyde and permeabilized for 15 min with PBS 0.1% Triton X100. The proximity ligations were performed according to the manufacturer's protocol. Preparations were mounted and analyzed similarly to immunofluorescence.

**Flow cytometry (FACS) analysis**. Flow cytometry experiments were performed on CytoFLEX flow cytometer (Beckman Coulter) instrument. After harvesting, suspensions of murine bone marrow and spleen were stained with 2 μl of CD45-APCH7 (BD Biosciences; 641417), 2 μl of CD33-PE (BD Biosciences; 555450) for 20 min, and cells were washed in PBS and resuspended in Annexin-V binding buffer (BD biosciences; 556454) plus 2 μl of Annexin-V-V500 (BD Biosciences; 561501). Absolute cell numbers of viable human blasts (CD45+/CD33+/Annexin-V−) were quantified using CountBright™ (Invitrogen). FACS analysis was also used to determine the mitochondrial mass of AML cells. Cells were washed with PBS and stained with MitoTracker™ Green FM (MTG, 1/10,000) for 20 min at 37 °C. Cells were then washed and resuspended in Annexin-V binding buffer (BD biosciences; 556454) plus 2 μl of Annexin-V-V500 (BD Biosciences; 561501).

Cell lipid content was measured by flow cytometry using the Bodipy 493/503 probe. Briefly, cells were washed, incubated with Bodipy (0.5 μg/mL) for 30 min at 37 °C, then washed and resuspended with Annexin binding buffer (BD bioscience; 556454) and stained with Annexin-V APC (BD bioscience; 550474) to exclude apoptotic cells. Data were analyzed with FlowJo v10 software (Tree Star Inc., Ashland, OR, USA).

**Cyto-ID®**. Autophagic flux was assessed using the Cyto-ID®-based procedure according to the manufacturer's instructions (Enzo Life Sciences, Switzerland). The fluorescence of the Cyto-ID® dye incorporated into the different AML cells was analyzed by a fluorescence-activated cell sorter on a Macsquant (Miltenyi Biotec, Paris, France) flow cytometer.

**Detection of lipid droplets by microscopy analysis**. Lipid droplets were stained using the Bodipy 493/503 probe. After cell plating on glass coverslips and fixation (see above), cells were washed and incubated with 1 µg/mL of Bodipy 493/503 diluted in 150 mM NaCl for 10 min at room temperature. Cells were then washed and slides mounted with ProLong™ Gold antifade medium with DAPI (Invitrogen). The number and the area of Bodipy dots per cell of at least 100 cells per independent experiment were determined with Image J software.

**Transmission electron microscopy–ultrastructural analysis**. Cell pellets were fixed with 2.5% glutaraldehyde in 0.1 M cacodylate buffer (pH 7.4) for 4 h. They were then postfixed with 1% osmium tetroxide in 0.1 M cacodylate buffer for 1 h, dehydrated in graded ethanols, and embedded in epon-araldite mixture. Ultrathin sections were stained with uranyl acetate and Reynold's lead citrate, and examined in a Hitachi HT 7700 transmission electron microscope (Hitachi High-Technologies, Tokyo, Japan) at 80 keV with acquisition performed on a CCD AMT XR41 camera at the CMEAB (Toulouse) or in an FEI Tecnai G2 at 200 KeV with acquisition performed on a Veleta camera (Olympus, Japan) at the PiCSL-FBI core facility (IBDM, AMU-Marseille).

Measurements of MERCs were performed using the Hitachi EMIP (Electron Microscopy Integrated Image Processing) software on micrographs (×6000–8000 magnification) acquired with the Hitachi HT 7700 transmission electron microscope.

**siRNA transfection**. The transfection of siRNA into MOLM14 cells was carried out using the Neon transfection system (Life technologies), according to the manufacturer's recommendations. Briefly, 70 nM of siRNA was used per condition. RNA interference-mediated gene knockdown was achieved using prevalidated Qiagen siRNAs (5′ > 3′) for Mtfn2 (Ref SI04375406; AAGACTATAAGCTGC GAATTA), Beclin1 (Ref SI00055587; TGGACAGTTTGGCACAATCAA), Dharmacon siRNAs (5′ > 3′) SMARTpool for IP3R1 (Ref L-006207-00-0010; UGGAA AGUCUGACCGAAUA, GCACAACAUCUACAUAUUA, GUAAGAAUGUCU ACACAGA, GCAAUCACAUGUGGAAAUU), and Dharmacon siGENOME Control Pool Non-Targeting #2 (Ref D-001206-14-20; UAAGGCUAUGAAGAG AUAC, AUGUAUUGGCCUGUAUUAG, AUGAACGUGAAUUGCUCAA, UGGUUUACAUGUCGACUAA).

**Plasmids and lentiviral infection**. plKO vectors containing the following shRNA sequences were used (5′ > 3′): shRNA Ctrl, purchased from Sigma (Ref HC016 MISSION pLKO.1-puro Non-Target shRNA Control; CGGGCGCGATAGCG CTAATAATTTCTCGAGAAATTATTAGCGCTATCGCGCTTTTTG); shRNA ATG12, purchased from Dharmacon (Ref RHS4696-20075354; CCGGGGACTCA TTGACTTCATCACTCGAGTGATGAAGTCAATGAGTCC); and shRNA VDAC1, purchased from Sigma (Ref SHCLNG-NM_003374; CCGGG CTTGGTCTAGGACTGGAATTCTCGAGAATTCCAGTCCTAGACCAAGCTT TTTG). The outer mitochondrial membrane–ER organelle linker and its control sequence, mAKAP1-mRFP-yUBC6 and mAKAP1-RFP, respectively (named OMM-ER and OMM, respectively) were a gift from Prof. György Hajnóczky (Thomas Jefferson University, Philadelphia). These sequences were added into an inducible lentiviral vector: Pinducer21.

Lentiviral transduction was carried out with lentiviral vectors (6 µg), 20 µL of lipofectamine 2000, psPax2 (4 µg, provides packaging proteins), and pMD2.G (2 µg, provides VSV-g envelope protein) plasmids into 293 T cells to produce lentiviral particles. Then, 72 h after cell transfection, 2 mL of the supernatants containing the virus particles were collected and added to $2 × 10^6$ MOLM14 cells in a six-well plate. Polybrene (8 µg/mL) was added and spinoculation was performed by centrifuging cells for 60 min at $800 × g$. Six hours after transduction, the medium containing the viruses was removed. Then, after an additional 72 h, transduced cells were selected with 1 µg/mL puromycin. All shRNA experiments were performed on the cell bulk, treated or not for 72 h with 1 µg/mL doxycycline for shRNA induction.

**Determination of FAO**. MOLM14 and U937 cell lines were beforehand treated or not with metformin (10 mM) or AA (10 µM). After 48 h of treatment, cells were incubated in duplicate for 3 h/37 °C with [1-$^{14}$C] palmitate (0.1 mCi/mL; PerkinElmer, Boston, MA) and cold palmitate (Sigma; 80 µM). Incubation buffer also contained 125 mM NaCl, 5 mM KCl, 2 mM CaCl$_2$, 1.25 mM KH$_2$PO$_4$, 1.25 mM MgSO$_4$, 25 mM NaHCO$_3$, 10 mM L-Carnitine, and 3% fatty acid-free BSA (Sigma). After incubation, FAO was measured by $^{14}$CO$_2$ trapped in 300 µL of benzethonium hydroxide 1 M (Sigma). Radioactivity of $^{14}$CO$_2$ was determined by liquid scintillation counting and normalized for cell number.

**Quantification of the lipogenesis**. AML cells were grown in six-well plates and treated with 10 mM metformin for 24 h. The cultures were then incubated with [$^3$H] acetate (0.2 µCi/mL) for 1 h. The incorporation of [$^3$H] acetate into the lipids was measured at the end of the incubation after four washes with cold PBS. Cellular lipids were extracted with a mixture of Butyl-PBD-Toluene[29,55,56]. The radioactivity in the organic phase was determined in a liquid scintillation counter. The counts per minute were normalized to the protein content in the total cell lysate.

**Lipidomic analysis**. Total lipid extracts were prepared following Bligh and Dyer procedures, then analyzed by gas chromatography/mass spectrometry for neutral lipid relative quantification[57]. Briefly, we evaluated the ratio of triglycerides on the total amount of neutral lipids, meaning the proportion of triglycerides among all the neutral lipids measured: total triglycerides (C49-TG + C51-TG + C53-TG + C55-TG + C57-TG + C59-TG) + total cholesterol ester (Chlo-C16 + Chlo-C18 + Chol-C20:4).

**Measurement of oxygen consumption in AML cultured cells**. OCR was measured at 37 °C using an XF24 extracellular flux analyzer (Seahorse Bioscience, Billerica, MA, USA). The day before the assay, the sensor cartridge was placed into the calibration buffer medium supplied by Seahorse Biosciences to hydrate overnight. Seahorse XFp microplates wells were coated with 25 µl of Cell-Tak (Corning; 354240) solution at a concentration of 22.4 µg/mL and kept at 4 °C overnight. A total of $2 × 10^5$ cells per well were adhered to Seahorse microplates and rested for 1 h in XF base minimal DMEM media containing 11 mM glucose, 1 mM pyruvate, and 2 mM glutamine before rates of glycolysis and oxidative phosphorylation were determined. Initially, baseline cellular OCR is measured, from which basal respiration can be derived by subtracting non-mitochondrial respiration. Next oligomycin, a complex V inhibitor, is added and the resulting OCR is used to derive ATP-linked respiration (by subtracting the oligomycin rate from baseline cellular OCR) and proton leak respiration (by subtracting non-mitochondrial respiration from the oligomycin rate). Next carbonyl cyanide-p-trifluoromethoxy-phenyl-hydrazon (FCCP), a protonophore, is added to collapse the inner membrane gradient, allowing the ETC to function at its maximal rate, and maximal respiratory capacity is derived by subtracting non-mitochondrial respiration from the FCCP rate. Lastly, AA and rotenone, inhibitors of complex III and I, are added to shut down ETC function, revealing the non-mitochondrial respiration. To determine the part of the OCR dependant on FAO, 3 µM of Etx was added to wells after 45 min of resting and 15 min before the OCR measurement. For oleate-induced OCR, cells were preincubated in Hank's Balanced Salt Solution (HBSS) supplemented with 20 µM of BSA or 20 µM of oleate-BSA for 1 h; subsequently, a readout of basal OCR was measured.

**Transcriptomic analysis**. Transcriptome profiling assays on MOLM14 and U937 cells untreated or treated with 10 mM metformin 24 h in independent triplicates were performed using the Affymetrix HuGene_2.0_starrays (GSE97346)[18]. RNA-sequencing subjected to differential expression analysis with a minimum of adjusted $p$-value < 0.05, absolute log2 fold change > 0.5. The 64 common significantly downregulated genes in MOLM14 (221 genes) and U937 (636 genes) were analyzed using Genomatix software (Genome Analyzer bioinformatics tool, www.genomatix.de, Munich, Germany) for further functional analyses (GO term) based on the Genomatix literature mining.

GSEA was performed using GSEA v3.0 tool developed by the Broad Institute. The enrichment scores were computed for the ranked genes from the metformin screen. Following parameters were used: number of permutations = 1000, permutation type = gene_set. Other parameters were left at default values.

**Tumor xenografts into NSG mice**. Animals were used for transplantation of AML cell lines in accordance with a protocol reviewed and approved by the Institutional Animal Care and Use Committee of Région Midi-Pyrénées (France). *NOD/LtSz-SCID/IL-2Rγchain null* (NSG) mice were produced at the Genotoul Anexplo platform at Toulouse (France) using breeders obtained from Charles River Laboratories. Mice were housed in sterile conditions using HEPA-filtered microisolators, and fed with irradiated food and sterile water. Eight-week-old mice were sublethally treated with busulfan (20 mg/kg) 24 h before injection of AML cells. AML cells (MOLM14, MOLM14 shARN control, or MOLM14 shARN VDAC1) were washed in PBS, and suspended in HBSS at a final concentration of $2 × 10^6$ cells per 200 µL of HBSS per mouse for tail vein injection. Mice injected with MOLM14 cells were treated from day-3 post cell injection to day-17, every day by oral gavage with a vehicle or with 1.5 mg/kg of IACS-010759 treatment. IACS-10759 was solubilized in water containing 0.5% methylcellulose before administration to mice. IACS-10759 was kindly provided by Dr. M. Konoplova and J. Marszalek. All mice were sacrificed after 17 days to harvest human leukemic cells from murine bone marrow and spleen. Engraftment in bone marrow and spleen was analyzed using flow cytometry. Mice survival time was also determined. Survival significance was determined by Kaplan–Meier curve and log-rank test.

**Software**. GSEA 3.0 software (http://software.broadinstitute.org/gsea/index.jsp), GeneSys software (https://www.syngene.com/support/software-downloads/), Zen Black2012 SP2 (https://www.zeiss.fr/microscopie/produits/microscope-software/zen-lite/zen-2-lite-download.html), Prism 6.0 software (https://www.graphpad.com/scientific-software/prism/), ImageJ 2.0 software (https://imagej.nih.gov/ij/download.html), GeneTool software (https://www.syngene.com/support/software-downloads/), Genomatix software (https://www.genomatix.de/), FlowJo 10.4.2 software (https://www.flowjo.com/solutions/flowjo/downloads), and Wave Desktop 2.6 (https://www.agilent.com/en/products/cell-analysis/software-download-for-wave-desktop).

**Statistical analysis**. Data from at least three independent experiments are reported as means ± standard error of the mean. Unless otherwise indicated, unpaired two-tailed Student's *t*-tests or paired *t*-tests for patients' studies analysis were carried out with Prism 8 software (GraphPad Software Inc). The Kaplan–Meier method was used to estimate leukemia-free survival in xenografted mice. Log-rank *p*-values (Mantel–Cox test) were used for comparisons of leukemia-free survival. For Genome Analyzer bioinformatics tool (Genomatix) the significance of the association between each GO term list was measured by Fisher's exact test.

**Reporting summary**. Further information on research design is available in the Nature Research Reporting Summary linked to this article.

## Data availability

Transcriptome profiling assays on MOLM14 and U937 treated or not with 10 mM metformin in independent triplicates. GEO: GSE97346. Source data are provided with this paper for Figs. 1–6 and Supplementary Figs. 1–8. Raw data are available on reasonable request.

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

## Acknowledgements

We are very grateful to P. Codogno and M. Selak for critically reading the manuscript. We thank Lucille Stuani, S. Giuriato, A. Bouloumié, O. Joffre, and A. Mairal for their intellectual/technical contributions. We are grateful to Manon Farcé and Laetitia Ligat for assistance with flow cytometry and microscopy, respectively. We thank all members of ANEXPLO Genotoul core facility (UMS006) for their support on mice experiments, the members of the CMEAB (Toulouse) for their help in electron microscopy acquisitions and the PiCSL-FBI core facilty (IBDM, AMU-Marseille), and member of the France-BioImaging national research infrastructure where a part of electron microscopy experiments was performed. We thank Prof. Marina Konopleva, Christopher Vellano, and Joseph Marszalek (The University of Texas MD Anderson Cancer Center) for providing IACS-010759 compound, Isabelle Bardey (EFS de Besançon) for normal hematopoietic cells, Prof. György Hajnóczky for giving us the sequences coding for the organelle linker mAKAP1-mRFP-yUBC6, and the control sequence mAKAP1-RFPn, and Prof. Eric Delabesse and Véronique De Mas for the management of the Biobank BRC-HIMIP (Biological Resources Centres-Inserm Midi-Pyrénées "Cytothèque des hémopathies malignes"). This work was supported by the ANR LABEX TOUCAN, the Cancéropole GSO, la Ligue régionale contre le Cancer, and la Ligue nationale contre le Cancer. We are grateful to our healthcare professionals for their boundless investment during the COVID-19 crisis.

## Author contributions

C.B., N.B., C.J., and J.-E.S. conceived and designed the experiments. S.S. made preliminaries experiments. C.B., N.B., and C.J. carried out most experiments. E.S. and T.F. performed Bodipy 493/503 staining and analysis by flow cytometry. M.F. carried out most of electron microscopy studies. Electron microscopy studies on shVDAC cells were performed by R.M. and A.C. C.L. and J.T. synthetized the outer mitochondrial membrane–ER organelle linker and its control sequence, mAKAP1-mRFP-yUBC6, and mAKAP1-RFP, respectively. N.E. and J.-C.P. gave us umbilical cord blood samples. C.C. performed Western blots experiments. A.B. and J.B.-M. evaluated the level of triglycerides. F.B. established the protocol for lipogenesis quantification. C.J., J.-E.S., N.B., C.B., E.S., T.F., S.M., C.R., S.B., and V.M.-D.M. analyzed and interpreted the data. C.J. and J.-E.S. wrote the manuscript.

## Competing interests

The authors declare no competing interests.
