## [Peer Review File · Nature Communications]

Reviewers' Comments:

Reviewer #1:

Remarks to the Author:

In this manuscript, Claudy Bosc et al. report that autophagy is an essential mechanism in AML cells regulating FFA availability to support mitochondrial oxidative metabolism in order to control cell proliferation and tumorigenicity. They further propose that the control of autophagy by mitochondria function is mediated through the regulation of contact sites between ER and mitochondria (called MERCs).

Overall the topic is relevant and timely, the study is well conducted and written and the results are interesting. Nevertheless, some points need to be addressed before further considerations.

Major points:

1. Using pharmacological approaches, the authors nicely reported that FFA are a major substrate in AML cells and that autophagy regulated lipid metabolism specifically in these cells, compared to healthy cells. They further used genetic approaches to modulate autophagy and confirmed the results observed with pharmacological inhibitors in cell lines. However, the authors need to confirm that silencing strategies used in MOLM14 cells (shATG12, siBec) have no impact in primary healthy cells.
2. The authors reported that inhibition of mitochondrial ETC reduced the % of mitochondria in contact with ER. What about MAM function? Is the exchange of phospholipids or calcium between both organelles reduced by metformin treatment?
3. The authors reported that alterations of MERCs occurred as early as 6h after mitochondrial inhibition and prior detection of autophagy inhibition. However, autophagy has been measured after 3h or 24h of inhibition of mitochondrial ETC, and therefore there is a gap where we don't know what really happens for autophagy! Therefore, the authors have to measure autophagy after 6h of metformin and antimycin A treatments in order to confirm that alterations of MERCs occurs before reduction of autophagy.
4. In parallel to TEM analysis, the authors used in situ PLA to analyse MERCs amount. It is unclear why VDAC1:IP3R1 dots are localized only on nucleus and not through the cytoplasm! In addition, this analysis is pertinent only whether the expression of both proteins is not regulated by treatment. Is the expression of VDAC1 and IP3R1 regulated at MERC interface in treated cells?
5. In order to experimentally reduce ER-mitochondria interactions, the authors performed VDAC1 or mitofusin 2 silencing. Both approaches target mitochondrial proteins that have other cellular functions outside of MERCs and therefore we cannot exclude that observed effects after the silencing of both proteins are related to alterations in mitochondria function rather than to MERC alteration! Therefore, in order to confirm their results, the authors have to silence a protein of MERCs rather located on ER side (i.e. IP3R1, as performed in Arruda et al Nat Med 2014) or to find another strategy specific of MERCs (i.e. FATE1, as performed in Tubbs et al. Diabetes 2019).
6. Another point related to these experiments of MERCs modulation, is the controversy related to mitofusin 2. Indeed, some authors proposed that mitofusin 2 acted as a tether between ER and mitochondria (de Britto OM et al. Nature 2008, Naon D et al. PNAS 2016) whereas others reported that mitofusin 2 would be a spacer (Cosson P et al. PLoS One 2012, Filadi R et al. PNAS 2015). Particularly, the controversy started from a TEM analysis compared to immunostaining approaches (Cosson P et al. PLoS One 2012). In their study, the authors used in situ PLA to confirm that silencing of mitofusin 2 reduced ER-mitochondria interactions. The analysis of MERCs by TEM in mitofusin 2-silenced cells should help to clarify this controversy.

7. The authors claim that inhibition of autophagy and subsequent accumulation of lipid droplets in metformin-treated cells is due to the loss of MERCs. Taking into account the point 5, this conclusion is actually not supported by results of the authors. One elegant approach to confirm this assumption should be to reinforce MERCs, using an organelle linker (as performed by Csordas G et al. JCB 2006), and to demonstrate that it prevents autophagy inhibition and lipid accumulation induced by metformin treatment.

8. Lastly, the authors also claim that autophagy occurs at MERCs, however no data support that proteins of autophagosome formation are present at MERC interface in AML cells and whether their location are regulated during inhibition of mitochondrial ETC. For that, the authors have to perform subcellular fractionation and analyse protein content by Western Blot. It should be interesting to test the presence of some lipid transfer proteins in MERC fractions!

Minor points:

1. The authors used metformin and antimycin A as pharmacological strategies to inhibit mitochondrial ETC. However, they need to confirm that mitochondrial respiration is well inhibited in these conditions.

2. From transmission electronic microscopy analysis, it is unclear how are measured MERCs. What is the minimal and maximal distances taken into account to consider that apposition between both organelle is a real contact site? How many cells are analysed and how many replicates were performed? Is the length of ER-mitochondria contact reduced or only the number?

3. Could this process of autophagy to supply FFA for mitochondrial oxidative metabolism be relevant in other cancer cells?

Reviewer #2:

Remarks to the Author:

The manuscript studies a link between lipophagy and mitochondria-ER contact sites in AML cells. Majority of the findings in the study are known – inhibition of autophagy leads to LD accumulation, decreased FAO, and decreased mitochondrial respiration. Conversely, inhibition of oxidative phosphorylation results in inhibition of lipophagy and increased LD. The only new concept provided by the authors is that blocking mitochondria-ER contact sites phenocopies oxidative phosphorylation inhibition in the context of LD, lipophagy and AML energetics.

However, whether the negative effect of mitochondria-ER contact sites disruption on AML growth is directly related to lipophagy is not clear. Disruption of mitochondria-ER contact sites will conceivably have several consequences on mitochondria fusion/fission, mitochondria function, metabolites feeding in to the mitochondria, and not necessarily limited to FA. Even if the authors conclusion/ extrapolation is true, it is difficult to envision how FAs resulting from lipophagy in the cytoplasm will be involved with the MERCs and not enter the mitochondria directly from the cytoplasm or from lysosome-mitochondria contact sites. To be able to connect the lipophagy defect to MERCs, the authors have to perform additional experiments to answer – whether LD biogenesis is affected with MERC inhibition for one, among others. The LD effect could be secondary to the changes in the mitochondria function upon MERC disruption.

PBMCs is not an appropriate control cell line to AML cells. The reviewer isn't clear about which AML cell type the authors are using. Are these from peripheral blood or derived from bone marrow? Notably, the authors make a summarizing statement, "this process is not implicated in healthy cells where lipids do not represent a major respiratory substrate" – this can be misread and misinterpreted as autophagy not being involved in lipid catabolism (lipophagy) in ANY healthy cell – which isn't the case because thousands of citations show the importance of lipophagy in different

healthy cell types. The authors also refer to PBMCs as normal hematopoietic cells, which in this reviewer's opinion isn't the case.

LC3 flux assay: The evaluation is wrong. The authors are directed to PMID 26799652 for the calculations. In short, normalized LC3 II/Actin densitometric values between +/- chloroquine are to be subtracted to get net LC3 flux. The authors will have 6 histograms for Figure 3a and 6 histograms for Figure 3b. The statistics will be applied between the LC3 flux of +/- Metformin (for Figure 3a) and the LC3 flux of +/- AA (for Figure 3b). The LC3 flux calculations need to be corrected in the supplementary figures as well. The corresponding IF for LC3 should also be performed +/- chloroquine.

In the schematic summary of the study (Figure 5d), the authors show the formation of autophagosomes from the ER. They don't include any data to show that ER is the exclusive source for autophagosome biogenesis in their cells. Additionally, inhibition of oxidative phosphorylation is shown to not form autophagosomes at all (from the ER), which isn't the case – as shown by the authors themselves. It is confusing to understand if the FFA are directly entering the mitochondria (as depicted pictorially by the authors), then what does the MERCs have to do with lipophagy? The disruption of MERCs and the effect on LD could be indirect.

Are the decreased MERCs due to decreased mitochondria number/mass in Metformin-treated AML cells? The EM images shown in Figure 2c seem to indicate that.

Figure 1g: image quality needs to be improved.

Figure S1f and S2f: images for healthy cells and BODIPY staining look very different.

Figure S1g bottom panel – please show one gel/ blot.

Figure S4g, S4h: Quantification is missing. Figure S4h is also missing the corresponding -CQ lanes for the two cell lines.

Reviewer #3:

Remarks to the Author:

This interesting and well written paper examines the role of autophagy and lipid metabolism in mitochondrial function (OxPhos) in acute myeloid leukemia cells. The likely source for the fatty acids is derived from autophagy. The authors go on to show that interaction between mitochondria and the endoplasmic reticulum, i.e., ER contact sites (MERC) likely plays an important role in this process. The authors use multiple approaches to document these outcomes. While the manuscript is very strong, some issues need clarification.

1. Fig 2D: Y-axis label is unclear (ratio to neutral lipids). Triglycerides are neutral lipids. What is measured in this analysis?
2. Method for fatty acid synthesis. The procedure, as written, does not measure fatty acid synthesis. Typically, cells are treated with labeled (3H or 14C)-acetate, cellular lipid is extracted with chloroform (or chloroform-methanol). Lipid appears in the organic extract and is quantified by beta scintillation counting and expressed per mg protein or total cells.
3. Is it correct that the source of the lipid forming lipid droplets is the result of autophagy involving lipase-mediated destruction of membrane lipids, release of non-esterified fatty acids and re-esterification of NEFA into neutral lipids? If so, are the neutral lipids only triacylglycerol or do they also include other neutral lipids, e.g., diacylglycerols and cholesterol esters.
4. If lipid droplets increase in response to metformin or actinomycin A, do proteins involved in lipid droplet organization increase, e.g., perilipins? The gene expression data (Fig 2a and b) only includes suppression of gene expression. Include data on genes that increase in response to

metformin/actimycin A treatment to accommodate increased lipid droplet formation.

4. Does inhibition of mitochondrial fatty acid oxidation increase the cellular capacity to synthesize and store lipids by increasing the expression of genes involved in these processes?

5. EM contact sites (MERC) are hard to see, increase magnification and improve resolution.

Reviewer #4:

Remarks to the Author:

Authors investigated the role of mitochondria respiratory chain inhibition in regulating autophagy and lipid metabolism in acute myeloid leukemia (AML) cells. They showed that inhibition of mitochondria ETC inhibited autophagy via decreased mitochondria-ER contact that resulting in accumulation of lipid droplet (LD) in AML cells. The study was well carried out and most data were of good quality. However, the concept that autophagy degrades lipid droplet (lipophagy) and released free fatty acids then burned via mitochondria beta oxidation has been known for decades and not novel. In addition, mitochondrial respiratory chain inhibitors modulate autophagy has been known for long time (PMID: 18032788; 22118681). Most data were descriptive and lack of novel mechanistic insight to advance on what we have known.

Specific comments:

1. Almost all the data were only shown in the cultured cells and only one piece of in vivo data in Figure 5c. The in vivo relevance of these findings were unclear as cells with knocking down of VDAC may have many other defects in addition to autophagy. Moreover, no autophagy data or lipid data were provided in the in vivo model.

2. Why the role of autophagy in regulating lipid metabolism is negligible in normal cells but not cancer cells?

3. Figure 2, Metformin has multiple targets such as acts as AMPK agonist or inhibition of MTOR to induce autophagy, which may be difficult to reconcile the accumulation of LDs in AML cells. These pathways have not been explored.

Point by point response to the referee comments

We thank all four reviewers for their constructive and thoughtful comments. In reply to their observations, we combined additional molecular, cellular and *in vivo* approaches to reinforce our conclusions. The data that we obtained markedly improved our model proposing that mitochondria activity regulates its supply of FFAs by regulating autophagosome formation at MERCs.

- We demonstrated that autophagosomes formation occurs at MERCs in AML cells using a challenging technical protocol of MERCs purification (**new Fig. 4c**).
- We solved the debate related to mitofusin 2 in our cell models by using electronic microscopy on AML cells transfected with an siRNA targeting mitofusin 2 that showed reduced MERCs compared to siRNA control cells (**new Fig. 5a,b**).
- Mechanistically, we strengthened the link between mitochondrial activity, MERCs, autophagy and lipid metabolism.
 - We first showed that mitochondria of AML cells pulsed with a fluorescent fatty acid lipid (RC₁₂) displayed less overlap with RC₁₂ when autophagy was inhibited (siBeclin1) or when MERCs were disrupted (siMtfn2) (**new Fig. 5i,j**), thus indicating that MERCs and autophagy are required to fuel mitochondrial metabolism and OxPHOS with lipids.
 - We then showed that the addition of exogenous fatty acids upon autophagy inhibition or MERCs disruption restored mitochondrial respiration (**new Fig. 5k**).
 - Finally, upon MERCs disruption, we performed the mirrored experiment by increasing the MERCs number using the organelle linker mAKAP1-mRFP-yUBC6. In AML cells with inhibited electron transport chain, restoring MERCs formation with this organelle linker prevented metformin-induced autophagy inhibition and lipid droplets accumulation (**new Fig. 5l-o**).
- Importantly, we now provided *in vivo* evidences for a role of the MERCs/fatty acid/autophagy axis in tumorigenicity of AML cells (**new Fig. 6c-i**).

Reviewer #1 (Remarks to the Author): Expertise in mitochondria-ER contact sites

In this manuscript, Claudy Bosc et al. report that autophagy is an essential mechanism in AML cells regulating FFA availability to support mitochondrial oxidative metabolism in order to control cell proliferation and tumorigenicity. They further propose that the control of autophagy by mitochondria function is mediated through the regulation of contact sites between ER and mitochondria (called MERCs).

Overall the topic is relevant and timely, the study is well conducted and written and the results are interesting. Nevertheless, some points need to be addressed before further considerations.

We thank reviewer#1 for his/her constructive and thoughtful comments that allowed us to significantly improve the quality of the manuscript.

Major points:

1. Using pharmacological approaches, the authors nicely reported that FFA are a major substrate in AML cells and that autophagy regulated lipid metabolism specifically in these cells, compared to healthy cells. They further used genetic approaches to modulate autophagy and confirmed the results observed with pharmacological inhibitors in cell lines. However, the authors need to confirm that silencing strategies used in MOLM14 cells (shATG12, siBec) have no impact in primary healthy cells.

As requested, we silenced Beclin1 with siRNA in primary blood cells from healthy donors (*i.e.* peripheral blood mononuclear cells from 5 healthy donors). Similarly to what we observed when using pharmacological inhibitors (Figures 1c, d and S1f), the genetic invalidation of autophagy had no significant impact on the number or the size of the lipid droplets (new panels in supplemental Fig. 1g-i, n=5). Moreover, as shown in new supplemental Figure 1q-r (n=4), while siBeclin 1 significantly decreased basal oxygen consumption rate (OCR) and ATP-linked OCR in AML cells (new Fig. 1k and new Supplemental Fig. 1o,p), it diminished these parameters in primary normal hematopoietic cells at much less extend.

Altogether, this additional set of data reinforces our observation that the involvement of autophagy in lipid catabolism and mitochondrial respiration is more limited in primary normal hematopoietic cells than in primary AML cells and cell lines.

2. The authors reported that inhibition of mitochondrial ETC reduced the % of mitochondria in contact with ER. What about MAM function? Is the exchange of phospholipids or calcium between both organelles reduced by metformin treatment?

We performed the RHOD2 assay to measure the mitochondrial calcium as a readout of MERCs function. The reduction of the percentage of mitochondria in contact with the ER upon ETC inhibition altered MAMs functions since metformin and antimycin A (AA) treatments significantly decreased mitochondrial calcium in the two AML cell lines that we used in this study (see new Supplemental Fig. 4e).

3. The authors reported that alterations of MERCs occurred as early as 6h after mitochondrial inhibition and prior detection of autophagy inhibition. However, autophagy has been measured after 3h or 24h of inhibition of mitochondrial ETC, and therefore there is a gap where we don't know what really happens for autophagy! Therefore, the authors have to measure autophagy after 6h of metformin and antimycin A treatments in order to confirm that alterations of MERCs occurs before reduction of autophagy.

We thank reviewer#1 for bringing up this important point. We additionally performed an analysis after 6h of metformin, antimycin A (new Fig. 3c,d and new Supplemental Fig. 3b,c) or IACS treatment (another inhibitor of the ETC complex I of the mitochondria, Attached Figure 1a), and we still did not observe a significant reduction in autophagy. Therefore, we confirmed that the alteration of MERCs occurred before the inhibition of autophagy.

4. In parallel to TEM analysis, the authors used in situ PLA to analyse MERCs amount. It is unclear why VDAC1:IP3R1 dots are localized only on nucleus and not through the cytoplasm! We apologize for not making this clear enough in the legends of the figures in the first version of the manuscript. We agree with reviewer#1 that VDAC1:IP3R1 dots appear to localize on

nucleus in these photos because the representative pictures of our PLA studies corresponded to orthogonal confocal projections of Z sections. Moreover, the lack of clarity in this observation might be amplified by the fact that AML cells are small non-adherent cells with a large nucleus and very small cytoplasm. This, PLA dots are throughout the cytoplasm and not in the nucleus. We now have indicated in the figure legend that the displayed pictures correspond to orthogonal confocal projections of Z sections.

In addition, this analysis is pertinent only whether the expression of both proteins is not regulated by treatment. Is the expression of VDAC1 and IP3R1 regulated at MERC interface in treated cells?

The reviewer is correct. We have now analyzed VDAC1 and IP3R1 expression upon 6, 24 and 48H of metformin and antimycin A treatment. As shown in the new Supplemental Fig. 4f, our western blot analysis did not reveal any impact of these two inhibitors on VDAC and IP3R1 expression.

5. In order to experimentally reduce ER-mitochondria interactions, the authors performed VDAC1 or mitofusin 2 silencing. Both approaches target mitochondrial proteins that have other cellular functions outside of MERCs and therefore we cannot exclude that observed effects after the silencing of both proteins are related to alterations in mitochondria function rather than to MERC alteration! Therefore, in order to confirm their results, the authors have to silence a protein of MERCs rather located on ER side (i.e. IP3R1, as performed in Arruda et al Nat Med 2014) or to find another strategy specific of MERCs (i.e. FATE1, as performed in Tubbs et al. Diabetes 2019).

As recommended by the reviewer, we silenced IP3R1 using siRNA technology. We obtained a good depletion of IP3R1 together with a marked reduction of LC3. This result indicates that the reduction of IP3R1 expression led to autophagy inhibition (Attached Figure 1d), such as observed with mitofusin2 depletion. At the discretion of the reviewer, this figure might be added as supplemental figure.

We would like to further stretching out that the depletion of these mitochondrial proteins did not alter mitochondria mass or number (new Supplemental Fig. 5b,c). We also confirmed that MERCs disruption after ETC inhibition occurred prior autophagy inhibition (see above point 3).

Therefore, along with the experiments performed with the ER-Mitochondria organelle linker (see below, point 7), the effects observed upon VDAC1 or mitofusin 2 silencing resulted from MERC alterations rather than from mitochondria dysfunction.

6. Another point related to these experiments of MERCs modulation, is the controversy related to mitofusin 2. Indeed, some authors proposed that mitofusin 2 acted as a tether between ER and mitochondria (de Britto OM et al. Nature 2008, Naon D et al. PNAS 2016) whereas others reported that mitofusin 2 would be a spacer (Cosson P et al. PLoS One 2012, Filadi R et al. PNAS 2015). Particularly, the controversy started from a TEM analysis compared to immunostaining approaches (Cosson P et al. PLoS One 2012). In their study, the authors used in situ PLA to confirm that silencing of mitofusin 2 reduced ER-mitochondria interactions. The analysis of MERCs by TEM in mitofusin 2-silenced cells should help to clarify this controversy.

We were also aware about this controversy related to mitofusin 2. Therefore, in addition to our PLA experiments that showed that mitofusin 2 depletion led to a reduced MERCs number in our model, we further performed transmission electron microscopy analysis on MOLM14 cells transfected with a control or mitofusin 2-specific siRNA. As in non-transfected MOLM14 cells (new Fig. 4a,b), numerous MERCs were observed in cells expressing the control siRNA (new Fig. 5a,b). In cells depleted for mitofusin 2, however, the percentage of mitochondria that interact with ER was reduced compared to control cells (new Fig. 5a,b). Moreover, as in metformin treated cells (new Fig. 2c) or in cells depleted for VDAC1 (new Fig. S5a), we observed an accumulation of lipid droplets in cells silenced for mitofusin 2. We are now convinced that this new EM study confirmed that mitofusin 2 silencing diminishes ER-mitochondria interactions in our cell model and answered the relevant question of the reviewer.

7. The authors claim that inhibition of autophagy and subsequent accumulation of lipid droplets in metformin-treated cells is due to the loss of MERCs. Taking into account the point 5, this conclusion is actually not supported by results of the authors. One elegant approach to confirm this assumption should be to reinforce MERCs, using an organelle linker (as performed by Csordas G et al. JCB 2006), and to demonstrate that it prevents autophagy inhibition and lipid accumulation induced by metformin treatment.

We acknowledge the reviewer#1 for this very good suggestion. We contacted Prof. György Hajnóczky who sent us two sequences, one coding for the organelle linker mAKAP1-mRFP-yUBC6, and the other one for the same molecule depleted from the ER targeting sequence (mAKAP1-RFP). Since AML cells are not easily transfectable, we synthesized the DNA sequences coding for the two peptides and we cloned them into an inducible lentiviral vector (Pinducer21) to infect MOLM14 cells. As expected, we observed a significant reduction of autophagy along with an accumulation of lipid droplets in metformin-treated MOLM14 cells expressing the control sequence (OMM) (new Fig. 5l-o). Conversely, restoring MERCs formation through the expression of the mAKAP1-mRFP-yUBC6 (i.e. OMM-ER) organelle linker (as functionally confirmed by the restoration of calcium uptake by the mitochondria; new Supplemental Fig. 5k), prevented metformin-induced autophagy inhibition and lipid droplet accumulation (new figure 5l-o).

These results strongly reinforce and support our conclusion that the inhibition of autophagy and the accumulation of lipid droplets observed upon metformin treatment are a consequence of MERCs disruption.

8. Lastly, the authors also claim that autophagy occurs at MERCs, however no data support that proteins of autophagosome formation are present at MERC interface in AML cells and whether their location are regulated during inhibition of mitochondrial ETC. For that, the authors have to perform subcellular fractionation and analyse protein content by Western Blot. It should be interesting to test the presence of some lipid transfer proteins in MERC fractions! We thank the reviewer for this very challenging but interesting crucial idea. We cited the work of T. Yoshimori (*Hamasaki et al, 2013, Nature*)¹ in the first version of our manuscript and gave him credit for experimentally showing that autophagosome formation occurs at ER-mitochondria contact sites in cancer cells. The involvement of MERCs in early events of autophagosomes formation has also been reported in fibroblasts (*Garofalo et al, 2016, Autophagy*)² and other recent studies reported a role for ER-mitochondria associations in

autophagy in HeLa and HEK293 cells (Hailey *et al*, 2010, *Nature*³; Wu *et al*, 2016, *Embo J*⁴; Gomez-Suaga, 2017, *Current Biology*⁵). Therefore, even if it is well established in the literature, we agree with reviewer#1 that the role of MERCs in autophagosome formation was not yet described in AML cells. So, to test the importance of MERCs in the autophagy of AML cells, we analysed the presence of key molecules involved in the early events of autophagic flux in isolated MERCs (New Figure 4c). This technically challenging experiment was adapted from the protocol established by Wieckowski *et al*⁶. Using 2.5 billion cells *per* experiment, we managed to isolate purified MERCs fraction with sufficient protein amount to perform Western blot analysis (new Fig 4c):

- FALC4, as a marker of MERCs, was enriched in the MERCs but not in the mitochondria or cytosol fractions.
- IP3R1, a marker of the ER, was not present in the MERCs fraction.
- VDAC1, a mitochondrial marker, was only enriched in the mitochondria fraction
- Actin as a cytosolic marker was absent in both MERCs and mitochondria fractions.

We then analysed the presence of key molecules involved in the early events of autophagic flux in the isolated MERCs fractions. Interestingly, we found that Vps34 and ATG16L, two proteins located on the isolation membrane (first step of autophagosomes formation), were constitutively present in these compartments (new Fig. 4c). These data strongly support that significant part of autophagosome formation occurs at the ER-mitochondria contact sites in AML cells.

Minor points:

1. The authors used metformin and antimycin A as pharmacological strategies to inhibit mitochondrial ETC. However, they need to confirm that mitochondrial respiration is well inhibited in these conditions.

We previously showed that ETC inhibitors (*i.e.* metformin and antimycin A) inhibit mitochondrial respiration and ETC activity in AML cell lines (Attached Figure 1b)^{7,8}. Nevertheless, we confirmed that both molecules strongly decreased mitochondrial respiration in our current cell models (new Fig. S2a)

2. From transmission electronic microscopy analysis, it is unclear how are measured MERCs. What is the minimal and maximal distances taken into account to consider that apposition between both organelle is a real contact site?

We apologise for not making this clear enough in the first version of the manuscript. We now describe in details how we quantified MERCs parameters (*i.e.* number, length and space between ER and mitochondria) in the method sections of our revised manuscript. Organelles juxtaposition was defined as a contact site when the distance between the two organelles was in the range of 10-100 nm. In average, the distance between ER and mitochondria was 43 nm +/- 5.8 in MOLM14 ctrl cells (analysed performed on 17 mitochondria) and 50nm +/- 4.3 in MOLM14 metformin treated cells (performed on 26 mitochondria). Similar results were obtained in U937 cells with the average distance between ER and mitochondria of 43 nm +/- 4.6 (30 mitochondria) in ctrl cells and of 42 nm +/- 5.7 in metformin treated cells (27 mitochondria).

How many cells are analysed and how many replicates were performed?

In the first version of the manuscript, we have quantified MERCs on:

- **9** non-treated MOLM14 cells and **14** MOLM14 cells treated with metformin
- **11** non-treated U937 cells and **15** U937 cells treated with metformin

In our revised manuscript, we have further measured MERCs on:

- **19** MOLM14 cells transfected with a siRNA control
- **21** MOLM14 cells transfected with a siRNA targeting mitofusin 2.

Is the length of ER-mitochondria contact reduced or only the number?

We thank the referee for helping us to sharpen our manuscript on this point. As suggested by the reviewer, we performed new measurements to evaluate the length of ER-mitochondria contact sites upon metformin treatment. Metformin reduced the number of the contact but without altering the length or the distance between both organelles (see above).

MERCs length in:

-MOLM14 + vehicle:	204 nm +/- 23
-MOLM14 + metformin:	228 nm +/- 20
-U937 + vehicle:	361 nm +/- 36
-U937 + metformin:	253 nm +/- 27

3. Could this process of autophagy to supply FFA for mitochondrial oxidative metabolism be relevant in other cancer cells?

It is now well established that cancer cells usually exhibit an exacerbated metabolism compared to their normal counterpart. Even if this metabolic reprogramming, known as the Warburg effect, consists originally to mainly use the glycolysis to produce ATP, a vast majority of cancer cells rely rather on mitochondrial oxidative metabolism⁹. Then, investigating the role of autophagy in supplying free fatty acids for mitochondrial oxidative metabolism in other cancer cells is more than relevant. Furthermore, several solid and blood cancers have dependency on OxPHOS and FAO pathways either in steady state (*Viale et al. Nature, 2014*¹⁰; *Carracedo et al. Nature Rev Cancer, 2013*¹¹; *Sawyer et al. Mol. Cancer Res. 2020*¹²) and in metastasis (*Pascual et al. Nature. 2017*¹³; *Lee et al. Science, 2019*¹⁴) but also upon treatment (*Samudio et al. JCI, 2010*¹⁵; *Farge et al. Cancer Discov*⁸, 2017), indicating that the better understanding of the contribution of respiratory sources especially FFA in mitochondrial OxPHOS might be crucial to deeper fight against cancer, drug resistance and relapse. In this purpose, AML seems to be a good model to study this concept. However, we hope that reviewer #1 agrees with us that these questions are most appropriately to be addressed in future studies.

Reviewer #2 (Remarks to the Author): Expertise in autophagy and lipid metabolism

The manuscript studies a link between lipophagy and mitochondria-ER contact sites in AML cells. Majority of the findings in the study are known – inhibition of autophagy leads to LD accumulation, decreased FAO, and decreased mitochondrial respiration. Conversely, inhibition of oxidative phosphorylation results in inhibition of lipophagy and increased LD. The only new concept provided by the authors is that blocking mitochondria-ER contact sites phenocopies oxidative phosphorylation inhibition in the context of LD, lipophagy and AML energetics. While we agree with reviewer#2 that some of our findings have already been described in other cell types, the fact that inhibition of autophagy leads to LD accumulation, decreased FAO, and decreased mitochondrial respiration has never been reported before in AML cells. Furthermore, our study has exhaustively reported for the first time in the same cellular model the entire mechanism of the interplay between ETCs, MAMs/MERCs, autophagosome, LD, lipophagy, FAO and OxPHOS activity, and the impact of this metabolic loop on cell proliferation (see new schematic diagram in new Fig. 6j).

We also agree that ETC inhibitors including metformin are known to regulate autophagy. However, while metformin through its agonist action on AMPK is mainly described as an autophagy inducer (*Sui X et al, Mol Pharm 2015*¹⁶, *Viollet et al, Front Biosci, 2009*¹⁷), we reported here together with recent studies the unexpected finding that OxPHOS inhibitors (metformin but also antimycin A and IACS-010759) can interfere and inhibit with autophagosome formation at MERCs. Moreover, as opposed to what it was shown in adipocytes and hepatocytes, we showed that metformin inhibits lipid degradation and FAO in AML cells (while this occurred at less extent in normal hematopoietic cells).

In conclusion, we demonstrated in this study using OxPHOS inhibitors as autophagy modulators for the first time a new regulatory loop in which mitochondria control its own supply of energy through the regulation of autophagosome formation at MERCs. To the best of our knowledge, we are convinced that this is an important and novel discovery in our field.

However, whether the negative effect of mitochondria-ER contact sites disruption on AML growth is directly related to lipophagy is not clear. Disruption of mitochondria-ER contact sites will conceivably have several consequences on mitochondria fusion/fission, mitochondria function, metabolites feeding in to the mitochondria, and not necessarily limited to FA.

To fully address this very relevant comment, we further experimentally strengthened the link between MERCs disruption, autophagy inhibition and lipid droplet accumulation as highlighted below:

- First, mitochondria mass or number was not affected by MERCs disruption (new Supplemental Fig. 4c,d), indicating that mitochondria morphology and biogenesis were not significantly impacted. Only MERCs functions were altered, since calcium uptake by the mitochondria was significantly decreased upon mitofusin 2 depletion (new Supplemental Fig 4e).
- Then, we pulsed cells with a fluorescent fatty acid lipid (RC₁₂) and studied its confocal overlap with mitochondria. When autophagy was inhibited (siRNA against Beclin1), or when MERCs were disrupted (siRNA against mitofusin2), mitochondria exhibited less overlap with RC₁₂. This result is likely due to the accumulation of RC₁₂ into the

cytoplasm (new Fig. 5i,j). Thus, MERCs and autophagy are required to fuel mitochondria with lipids.

- Moreover, the addition of fatty acids upon autophagy inhibition or MERCs disruption restored mitochondrial respiration (new Fig. 5k).
- Finally, after disrupting MERCs, we decided to perform the mirrored experiment by increasing the MERCs/MAMs number in cells. Therefore, we contacted Prof György Hajnóczky who generated an organelle linker used in several studies from his own group (*Csordas et al, JCB, 2006*¹⁸) and others (*Gomez-Suaga et al, Current Biology, 2017*⁵; *Basso et al, Pharmacological Research, 2018*¹⁹). He sent us two sequences, one encoding for the organelle linker mAKAP1-mRFP-yUBC6 (i.e. OMM-ER), and the other one for the same molecule depleted from the ER targeting sequence (mAKAP1-RFP). Since AML cells are not transfectable, we synthesized the two DNA sequences coding for the two peptides and we cloned them into an inducible lentiviral vector (Pinducer21) to infect MOLM14 cells. As expected, we observed a significant reduction of autophagy alongside with an accumulation of lipid droplets in metformin-treated MOLM14 cells expressing the control sequence (OMM), (new Figure 5l-o). In contrast, restoring MERCs formation through the expression of the organelle linker OMM-ER, confirmed by the restoration of calcium uptake by the mitochondria (new Supplemental Fig. 5k), prevented metformin-induced autophagy inhibition and lipid droplets accumulation (new Fig. 5l-o).

We believe that this new set of data would definitively convince the reviewer and reinforce that the inhibition of autophagy and the subsequent accumulation of lipid droplets in metformin-treated cells were due to the loss of MERCs.

Even if the authors conclusion/extrapolation is true, it is difficult to envision how FAs resulting from lipophagy in the cytoplasm will be involved with the MERCs and not enter the mitochondria directly from the cytoplasm or from lysosome-mitochondria contact sites.

To be able to connect the lipophagy defect to MERCs, the authors have to perform additional experiments to answer – whether LD biogenesis is affected with MERC inhibition for one, among others.

Our hypothesis was supported by i) the subcellular fractionation experiment showing that part of autophagosomes formation occurred at ER-mitochondria contact sites and by the fact that the defect in lipid degradation was due to a decrease in the number of MERCs (since MERCs disruption led to the accumulation of lipid droplet). To further connect MERCs to lipophagy, we followed the suggestion made by the reviewer#2 and we investigated whether lipid droplet biogenesis was affected by MERCs inhibition (*i.e.* upon mitofusin 2 depletion). We performed a nuclear/cytosol fractionation assay to monitor SREBP1/2 localisation. SREBP1 and 2 were found mainly in the nucleus of cells expressing or not mitofusin 2 (new Supplemental Fig. 5g). This result indicates that MERCs disruption does not impact SREBP1/2 subcellular localisation (and therefore lipid biogenesis). We also analysed the expression of several proteins implicated in lipid droplet biogenesis (*e.g.* ADRP, ACCLY, FASN) by western blot. We did not observe any changes in the expression of any of those proteins upon depletion of mitofusin 2 (new Supplemental Fig. 5f). Of note, metformin treatment markedly reduced lipogenesis (Supplemental Fig. 2i in first manuscript version or new Supplemental Fig. 2j) but did not

change SREPB1 and 2 subcellular localisations (new Supplemental Fig. 2l). Additionally, metformin and antimycin A treatments did not modify the level of expression of ADRP, ACCLY and FABP4, and only mildly increased expression level of FASN (new Supplemental Fig. 2k). These results support the hypothesis that the accumulation of lipid droplet observed upon ETC inhibition is due to a defect in degradation.

Together, these data argue for the fact that the disruption of MERCs does not alter lipid droplet biogenesis, and that lipid droplet accumulation is directly due to a defect in lipid degradation, thus linking lipophagy defect to MERCs.

The LD effect could be secondary to the changes in the mitochondria function upon MERC disruption.

We analysed mitochondria mass or number and showed that these features are not affected by MERCs disruption (new Supplemental Fig. 4c,d), strongly suggesting that mitochondria morphology and biogenesis are not impacted. Only MERCs functions seem to be altered since calcium uptake by the mitochondria was significantly decreased upon mitofusin 2 depletion (new Supplemental Figure 4e). Additionally, we confirmed that MERCs disruption after ETC inhibition (after 6 hours, new Fig.4e) occurred prior autophagy inhibition (24 hours, new Fig. 3a-d).

In conclusion, the experiments performed with the mitochondria-ER organelle linker (see above; new Fig. 5l-o) along with this new dataset strongly support that the effects observed upon VDAC1 or mitofusin 2 silencing result from MERC alteration rather than from mitochondrial dysfunction.

PBMCs is not an appropriate control cell line to AML cells.

The reviewer isn't clear about which AML cell type the authors are using. Are these from peripheral blood or derived from bone marrow?

Notably, the authors make a summarizing statement, "this process is not implicated in healthy cells where lipids do not represent a major respiratory substrate" – this can be misread and misinterpreted as autophagy not being involved in lipid catabolism (lipophagy) in ANY healthy cell – which isn't the case because thousands of citations show the importance of lipophagy in different healthy cell types. The authors also refer to PBMCs as normal hematopoietic cells, which in this reviewer's opinion isn't the case.

We agree with reviewer#2 that PBMCs are not the perfect control for AML cells. CD34 positive cells from normal BM could also represent a "normal" counterpart for AML cells. Therefore, we performed experiments on both cell types that probably represent the best control that could be used for primary AML cells. A new set of *in vitro* results with CD34 positive cells purified from cord blood rather than PBMC is now included in Fig. 1a,c,d; S1d; 2f,g; 3f. Results obtained with CD34 positive cells are similar to those found with PBMC. Therefore, we decided to combine in the same graph results obtained with PBMC and CD34 together under the name "healthy hematopoietic cells". However, the Attached Figure 2 showed data that we only got with CD34 positive cells. Consistent with PBMC's results, etomoxir slightly decreased OCR and ATP linked to respiration in CD34 positive cells (Attached Figure 2a,b). Therefore, FFA participate to fuel TCA cycle and oxidative phosphorylation in CD34 positive cells but to a less extend that in AML cells (new Fig. 1a, S1a,b,d). We further investigated the impact of

metformin on lipid metabolism and on autophagy. The inhibition of the ETCI in CD34 positive cells did not modify neither the number nor the area of lipid droplets (Attached Figure 2c,d) and had no significant effect on autophagy (Attached Figure 2e-g). We finally apologize for not having been clear enough that the AML cells are from peripheral blood. This is now indicated in the material and methods section.

Altogether, these results indicate that the oxidative phosphorylation regulate lipid metabolism through autophagy regulation in AML cells and to a less extend in healthy hematopoietic cells.

LC3 flux assay: The evaluation is wrong. The authors are directed to PMID 26799652 for the calculations. In short, normalized LC3 II/Actin densitometric values between +/- chloroquine are to be subtracted to get net LC3 flux. The authors will have 6 histograms for Figure 3a and 6 histograms for Figure 3b. The statistics will be applied between the LC3 flux of +/- Metformin (for Figure 3a) and the LC3 flux of +/- AA (for Figure 3b). The LC3 flux calculations need to be corrected in the supplementary figures as well. The corresponding IF for LC3 should also be performed +/- chloroquine.

We apologize for not having correctly evaluated autophagic flux in the first version of the manuscript. To fully investigate this point, we now performed all the calculations as recommended by reviewer#2 and detailed them in “Guidelines for the use and interpretation of assays for monitoring autophagy”. Normalized LC3 II/Actin densitometric values between +/- chloroquine were then subtracted to get net LC3 flux in new Figures 3a-f and in Figures S3a-d. Experimental and statistical results obtained with the correct analysis were comparable to those obtained in the previous version of the submitted manuscript, confirming our initial conclusion that ETC inhibition significantly decreased autophagy flux in AML cells. The corresponding LC3 IF done in AML cells lines +/- chloroquine are now also presented (new Fig. 3g,h and Fig. S3d,e).

In the schematic summary of the study (Figure 5d), the authors show the formation of autophagosomes from the ER. They don't include any data to show that ER is the exclusive source for autophagosome biogenesis in their cells.

We apologise for not making this clear enough in the first version of our schematic diagram. We thank the reviewer for this specific comment that helped us to improve our working model and conclusions. We never meant that ER was the exclusive source for autophagosome biogenesis. We wanted to illustrate that based on the literature (*Hamasaki et al, 2013, Nature; Garofalo et al, 2016, Autophagy; Hailey et al, 2010, Cell; Wu et al, 2016, Embo J; Gomez-Suaga, 2017, Current Biology*)¹⁻⁵, and now also based on our new subcellular fractionation data (new Figure 4c), a part of autophagosomes formation occurred at ER-mitochondria contact sites in AML cells. The graphical abstract has now been modified to better support our interpretation (new Fig. 6j).

Additionally, inhibition of oxidative phosphorylation is shown to not form autophagosomes at all (from the ER), which isn't the case as shown by the authors themselves.

The inhibition of oxidative phosphorylation decreased the autophagosomes formation but did not completely inhibit their biogenesis. Indeed, LC3B-II proteins could still be detected by western blot upon ETC inhibition + chloroquine (new Fig. 3a-f and S3a-c) and some autophagosomes are still present as depicted by IF studies (new Figure 3g,h and S3d-f).

Accordingly, we think that the new data shown in the new figure 4c reinforced our assumption that part of autophagosome formation occurs at the ER-mitochondria contact sites in AML cells. Therefore, ETC inhibition mainly targets autophagosomes biogenesis at this specific location by disrupting ER-mitochondria contact sites.

It is confusing to understand if the FFA are directly entering the mitochondria (as depicted pictorially by the authors), then what does the MERCs have to do with lipophagy? The disruption of MERCs and the effect on LD could be indirect.

We agree with reviewer#2 that the disruption of MERCs, the inhibition of autophagy and the defect on LD could be indirect. To experimentally and mechanistically improve the direct link between these three biological processes, we performed a series of new experiments (see detailed above: pages 7,8): pulse chase experiments with fluorescent lipid, restoration of full OCR with addition of lipids in cells in which autophagy was blocked or in which MERCs was disrupted, and the use of the organelle linker that prevents autophagy inhibition and lipid accumulation upon metformin treatment. Moreover, our subcellular fractionation to obtain purified MERCs, in which we observed the presence of proteins implicated in autophagosome formation (new Fig. 4c), further supports the existence of a direct link between MERCs and autophagy.

Are the decreased MERCs due to decreased mitochondria number/mass in Metformin-treated AML cells? The EM images shown in Figure 2c seem to indicate that.

Regarding mitochondria number/mass, we found that the number of mitochondria present on EM pictures of MOLM14 and U937 cells (**9** non-treated MOLM14 and **14** MOLM14 treated with metformin; **11** non-treated U937 and **15** U937 treated with metformin) was not modified upon metformin treatment (new Supplemental Fig. 4c), and that the mass of mitochondria analysed by FACS using mitotracker green probe was also not altered by metformin treatment (new Supplemental Fig. S4d).

Figure 1g: image quality needs to be improved.

We assume that reviewer 2 is referring to Figure 1e. As suggested by the reviewer, the source and quality of the immunofluorescence (IF) pictures shown in Figure 1e was re-examined and more representative pictures were included in the revised manuscript (see new Figure 1e).

Figure S1f and S2f: images for healthy cells and BODIPY staining look very different.

We agree with reviewer#2 that the BODIPY staining for healthy cells in Figure S2f looked more diffuse than in Figure S1f. It is likely due to the fact that there is usually less BODIPY dots in primary cells, we therefore included more representative pictures for healthy cells in new Supplemental Fig. 2g.

Figure S1g bottom panel – please show one gel/ blot.

We apologize for this error of labelling. It is now corrected in the new manuscript version (new Fig. S1g).

Figure S4g, S4h: Quantification is missing. Figure S4h is also missing the corresponding -CQ lanes for the two cell lines.

As requested, we are showing LC3BII/actin quantifications for new Supplemental Fig. 5d and 5e, and we are now including the corresponding lanes without chloroquine for the two cell lines (new Supplemental Fig. 5e).

Reviewer #3 (Remarks to the Author): Expertise in lipidomics and transcriptomics

This interesting and well written paper examines the role of autophagy and lipid metabolism in mitochondrial function (OxPhos) in acute myeloid leukemia cells. The likely source for the fatty acids is derived from autophagy. The authors go on to show that interaction between mitochondria and the endoplasmic reticulum, i.e., ER contact sites (MERC) likely plays an important role in this process. The authors use multiple approaches to document these outcomes. While the manuscript is very strong, some issues need clarification.

We would like to thank the reviewer#3 for pointing out that our paper is interesting and well written as well as for his/her constructive comments/questions that helped us to improve its clarity.

1. Fig 2D: Y-axis label is unclear (ratio to neutral lipids). Triglycerides are neutral lipids. What is measured in this analysis?

We apologise for not making this clear enough in the first version of the manuscript.

The Y-axis represents the ratio of triglycerides on the total amount of neutral lipids, meaning the proportion of triglycerides among all the neutral lipids measured: total triglycerides (C49-TG + C51-TG + C53-TG + C55-TG + C57-TG + C59-TG) + total cholesterol ester (Chlo-C16 + Chlo-C18 + Chol-C20:4). In the revised manuscript, we have stated this point in the experimental procedures.

2. Method for fatty acid synthesis. The procedure, as written, does not measure fatty acid synthesis. Typically, cells are treated with labeled (3H or 14C)-acetate, cellular lipid is extracted with chloroform (or chloroform-methanol). Lipid appears in the organic extract and is quantified by beta scintillation counting and expressed per mg protein or total cells.

The organic solvent used in our study in order to extract lipids was toluene. In this case it has the same function than chloroform as described in *Le Marchand-Brustel et al, Endocrinology, 1995*²⁰; *Loubière C et al, Oncotarget, 2015*²¹; *Regazzetti C et al, PLoS One. 2012*²². In order to clarify this point in the experimental section of the revised manuscript, we have changed the text as highlighted on pages 21 in the revised manuscript to: “*AML cells were grown in 6-well plates and treated with 10mM metformin for 24 h. The cultures were then incubated with [3H] acetate (0.2 µCi/ml) for 1h. The incorporation of [3H] acetate into the lipids was measured at the end of the incubation after four washes with cold PBS. Cellular lipids were extracted with a mixture of Butyl-PBD-Toluene^{22,36,37}. The radioactivity in the organic phase was determined in a liquid scintillation counter. The counts per minute (cpm) were normalized to the protein content in the total cell lysate*”.

3. Is it correct that the source of the lipid forming lipid droplets is the result of autophagy involving lipase-mediated destruction of membrane lipids, release of non-esterified fatty acids and re-esterification of NEFA into neutral lipids? If so, are the neutral lipids only triacylglycerol or do they also include other neutral lipids, e.g., diacylglycerols and cholesterol esters.

Autophagosomes engulf lipid droplets or part of lipid droplets and then their fusion with lysosomes that contain the lysosomal acid lipase, allows the degradation of lipid droplets composed of triglycerides and cholesterol esters. Lysosomal acid lipase hydrolyses triglycerides and cholesterol ester.

To confirm in our model that the source of lipids is the result, at least in part, of autophagy, we used a highly specific inhibitor of the lysosomal acid lipase Lalistat2²³ that is now commercially available, and tested its impact on lipid metabolism. Lalistat2 induced lipid droplet accumulation (new Supplemental Fig. 1j,k), indicating that the increased number of lipid droplets observed upon 3-MA treatment or Beclin 1 depletion is due to a blockage of autophagy. Afterwards, free fatty acids are released by the action of lipases on triglycerides and shuttled into mitochondrial matrix through carnitine system CPT1-CACT-CPT2. Then fatty acids are further degraded and oxidized into the mitochondria.

4. If lipid droplets increase in response to metformin or actimycin A, do proteins involved in lipid droplet organization increase, e.g., perilipins?

Our hypothesis is that metformin treatment led to the accumulation of lipid droplet because there is a defect in lipid degradation due to a decrease in the number of MERCs. MERCs are a location of autophagosome biogenesis. Therefore, if there is less MERCs, less autophagosomes will be formed and consequently less lipid droplets will be degraded.

Consistent with this hypothesis, we now showed that autophagosomes formation occurs at ER-mitochondria contact sites in AML cells (new Fig. 4c). Nevertheless, as recommended by reviewer#3, we investigated whether proteins involved in lipid droplet organization are increased. To do so, we performed a kinetic of metformin and antimycin A treatments, and analysed the level of expression of ADRP and FABP4 by western blot (of note, perilipin 1 and FATP were not detected). ETC inhibition does not modify the level of expression of ADRP and FABP4 (new Supplemental Fig 2k), supporting the notion that the lipid droplet accumulation observed upon ETC inhibition is due to a defect in lipid degradation. Similar results were obtained upon mitofusin 2 depletion (e.g. MERCs reduction) (new Supplemental Fig. 5f).

The gene expression data (Fig 2a and b) only includes suppression of gene expression. Include data on genes that increase in response to metformin/actimycin A treatment to accommodate increased lipid droplet formation.

As suggested, we now included all genes up-regulated upon metformin treatment in the new manuscript version (new Fig. 2a-b). In accordance with our analysis on protein expression, no gene expression involved in fusion/formation of lipid droplets was modified upon metformin treatment (FITM1/2; CIDE family with FSP27 or CIDEC, ADRP, Perilipins (1, 3, 4), Seipin). We also analysed more precisely the expression level of genes involved into:

- the synthesis of triglycerides: GPAT4, DGAT1 et DGAT2
- the trafficking of lipid droplets: ARF/COPI, proteins SNARE (SNAP23, syntaxin-5, VAMP-4, gamma synuclein), trans-golgi proteins (ARFRP1-ARL1 system, Rab18).
- and the transcription factors regulating lipid synthesis: SREPB1/2, PPAR and TFAP2.

Of note, none of these genes was impacted by metformin treatment. To strengthen this point, we then decided to perform a nuclear/cytosol fractionation assay to monitor SREBP1/2 localisation upon metformin treatment. ETC inhibition did not change SREPB1/2 subcellular localisation (new Supplemental Fig. 2l).

Altogether, these results reinforced our hypothesis that O_xPHOS/ETC inhibitor induced a defect in lipid degradation, leading to the accumulation of lipid droplets.

4. Does inhibition of mitochondrial fatty acid oxidation increase the cellular capacity to synthesize and store lipids by increasing the expression of genes involved in these processes?

As suggested by the reviewer, we investigated whether the inhibition of mitochondrial fatty acid oxidation with Etomoxir (an inhibitor of mitochondrial FA transporter/translocase CPT1) modulates the expression of genes involved in lipid synthesis and storage. Upon Etomoxir treatment (at low dose: 3 μ m), we determined the expression level of ACCLY, FASN, ACC, perilipins and FABP4 by western blot. The expression levels of proteins involved in lipid storage or transport (perilipins, FABP4) were not modified, whereas proteins implicated in fatty acid biosynthesis such as ACLY, ACC, FASN were down-regulated upon fatty acid oxidation inhibition (Attached Figure 3A). At the discretion of the reviewer, this figure can be added as supplemental figure. These data are in favour of a defect in lipid degradation rather than an increase of lipid synthesis.

5. EM contact sites (MERC) are hard to see, increase magnification and improve resolution.

As suggested by the reviewer, we have now included high resolution EM pictures and higher magnification to better visualize MERCs in the revised manuscript (see new Fig. 4a, S4a, 5a, S5a).

Reviewer #4 (Remarks to the Author): Expertise in autophagy

Authors investigated the role of mitochondria respiratory chain inhibition in regulating autophagy and lipid metabolism in acute myeloid leukemia (AML) cells. They showed that inhibition of mitochondria ETC inhibited autophagy via decreased mitochondria-ER contact that resulting in accumulation of lipid droplet (LD) in AML cells. The study was well carried out and most data were of good quality. However, the concept that autophagy degrades lipid droplet (lipophagy) and released free fatty acids then burned via mitochondria beta oxidation has been known for decades and not novel. In addition, mitochondrial respiratory chain inhibitors modulate autophagy has been known for long time (PMID: 18032788; 22118681). Most data were descriptive and lack of novel mechanistic insight to advance on what we have known.

We thank the reviewer#4 for pointing out that our work was well carried out with good quality. The reviewer#4 disappointment about our concept regarding the interconnection between mitochondria and autophagy through mitochondria-ER contacts sites strongly pushed us to improve the clarity of our conclusions, and to experimentally address key aspects of our novel concept (as developed/addressed in the paragraph below and highlighted in our revised manuscript and figures). We strongly hope that these additional/complementary experiments and works will definitively convince the reviewer.

We agree with the reviewer#4 that it was already established that autophagy could degrade lipid droplets²⁴ and release free fatty acids that are then burned *via* mitochondria beta-oxidation. We also agree that ETC inhibitors including metformin are known to regulate autophagy. However, while metformin through its agonist action on AMPK is mainly described as an autophagy inducer (*Sui X et al, Mol Pharm 2015, Viollet et al, Front Biosci, 2009*)^{16,17}, we reported here with recent studies the unexpected finding that OxPHOS inhibitors (metformin but also antimycin A and IACS010759) can interfere and inhibit with autophagosome formation at MERCs. Moreover, as opposed to what it was shown in adipocytes and hepatocytes, we showed that metformin inhibits lipid degradation and FAO in AML cells (while this occurred at less extent in normal hematopoietic cells).

In conclusion, we demonstrated in this study using OxPHOS inhibitors as autophagy modulators for the first time a new regulatory loop in which mitochondria control its own supply of energy through the regulation of autophagosome formation at MERCs. To the best of our knowledge, we are convinced that this is an important and novel discovery in our field.

Furthermore, to fix this lack of mechanistic insight, we further experimentally strengthened the link between MERCs disruption, autophagy inhibition and lipid droplet accumulation as highlighted below:

- First, mitochondria mass or number was not affected by MERCs disruption (new Supplemental Fig. 4c,d), indicating that mitochondria morphology and biogenesis were not significantly impacted. Only MERCs functions were altered, since calcium uptake by the mitochondria was significantly decreased upon mitofusin 2 depletion (new Supplemental Fig 4e).
- Then, we pulsed cells with a fluorescent fatty acid lipid (RC₁₂) and studied its confocal overlap with mitochondria. When autophagy was inhibited (siRNA against Beclin1), or when MERCs were disrupted (siRNA against mitofusin2), mitochondria exhibited less overlap with RC₁₂. This result is likely due to the accumulation of RC₁₂ into the

cytoplasm (new Fig. 5i,j). Thus, MERCs and autophagy are required to fuel mitochondria with lipids.

- Moreover, the addition of fatty acids upon autophagy inhibition or MERCs disruption restored mitochondrial respiration (new Fig. 5k).
- Finally, after disrupting MERCs, we decided to perform the mirrored experiment by increasing the MERCs/MAMs number in cells. Therefore, we contacted Prof György Hajnóczky who generated an organelle linker used in several studies from his own group (Csordas *et al*, *JCB*, 2006)¹⁸ and others (Gomez-Suaga *et al*, *Current Biology*, 2017; Basso *et al*, *Pharmacological Research*, 2018)^{5,19}. He sent us two sequences, one encoding for the organelle linker mAKAP1-mRFP-yUBC6 (i.e. OMM-ER), and the other one for the same molecule depleted from the ER targeting sequence (mAKAP1-RFP). Since AML cells are not transfectable, we synthesized the two DNA sequences coding for the two peptides and we cloned them into an inducible lentiviral vector (Pinducer21) to infect MOLM14 cells. As expected, we observed a significant reduction of autophagy alongside with an accumulation of lipid droplets in metformin-treated MOLM14 cells expressing the control sequence (OMM), (new Figure 5l-o). In contrast, restoring MERCs formation through the expression of the organelle linker OMM-ER, confirmed by the restoration of calcium uptake by the mitochondria (new Supplemental Fig. 5k), prevented metformin-induced autophagy inhibition and lipid droplets accumulation (new Fig. 5l-o).

We think that this new dataset reinforces that the inhibition of autophagy and the subsequent accumulation of lipid droplets in metformin-treated cells were due to the loss of MERCs, and might definitively convince the reviewer.

Specific comments:

1. Almost all the data were only shown in the cultured cells and only one piece of *in vivo* data in Figure 5c. The *in vivo* relevance of these findings were unclear as cells with knocking down of VDAC may have many other defects in addition to autophagy. Moreover, no autophagy data or lipid data were provided in the *in vivo* model.

We agree with the reviewer#4 that most of our work was done in cultured cells even though a substantial part was done on primary cells. To follow the constructive reviewer#4's recommendation, and to complete our *in vivo* study, we decided to directly target mitochondrial oxidative phosphorylation with a highly specific ETCI inhibitor IACS-010759 orally administrable in mice²⁵. This new ETCI inhibitor is actually under clinical trial phase 1 (NCT02882321). Drs M. Konopleva and J. Marszalek (MD Anderson Cancer Institute, USA) generously provided us this compound.

We first obtained the same *in vitro* results with this new drug than with metformin or antimycin A on autophagy and lipid metabolism. As expected, IACS-010759 treatment led to a marked accumulation of lipid droplets (new Supplemental Fig. 6a-b), concomitantly to a significant autophagy inhibition *in vitro* (new Supplemental Fig. 6c-f). Then, we daily treated NSG mice engrafted with MOLM14 with a low dose (1.5 mg/kg) of IACS-010759 for 14 days. Mice treated with IACS-010759 displayed a significant reduction of the total tumour burden and an increase in overall survival compared to mice treated with vehicle (new Fig. 6c,d). This confirmed that the tumorigenicity of AML cells is linked to the dependency to mitochondrial

metabolism and ETC1 activity. *Ex vivo* experiments performed on purified bone marrow cells from mice treated with IACS-010759 demonstrated an accumulation of bigger lipid droplets (new Fig. 6e,g), a marked reduction of autophagy (new Fig. 6h), and a reduced number of mitochondria-ER contact sites compared to cells from control mice (new Fig. 6i). Thus, the tumorigenicity of AML cells is further linked to the ability of their mitochondria to regulate their fatty acid supply by the regulation of autophagy through mitochondria-ER contact sites formation. This increased FFA required to maintain high level of oxidative phosphorylation (new Fig. 6j).

2. Why the role of autophagy in regulating lipid metabolism is negligible in normal cells but not cancer cells?

Compared with normal hematopoietic cells, AML cells have a lower spare reserve capacity in the respiratory chain, and different oxidative metabolism (*Sriskanthadevan S et al Blood 2015*)²⁶. AML cells also exhibit higher mitochondrial mass with a concomitant enhanced mitochondrial biogenesis, and an increased basal oxygen consumption, *versus* normal hematopoietic cells (*Skrtic et al. Cancer Cell, 2011*)²⁷. To support oxidative phosphorylation process, several respiratory substrates might be used and oxidized through their respective catabolic pathways by cells. Indeed, the major sources of NADH/FADH₂ for ETC are produced from the oxidation of glucose, glutamine and other amino acids, branched chain amino acids or fatty acids. Finally, AML OxPHOS and viability are more dependent on FAO (therefore cellular lipid degradation by lipolysis and/or lipophagy) than normal hematopoietic OxPHOS (*Samudio et al JCI. 2010*¹⁵; *Lee et al Cancer Res. 2015*)²⁸).

3. Figure 2, Metformin has multiple targets such as acts as AMPK agonist or inhibition of MTOR to induce autophagy, which may be difficult to reconcile the accumulation of LDs in AML cells. These pathways have not been explored.

In our first intention, we did not investigate whether AMPK was implicated in the observed metformin's effects since we previously showed that AMPK is transiently activated upon metformin treatment, and that the reduction of proliferation due to metformin treatment was independent of AMPK (*Scotland et al. Leukemia, 2013*)⁷. However, as suggested by the reviewer#4 we assessed whether AMPK was implicated in metformin-dependent autophagy inhibition or lipid accumulation. Using MOLM14 cells depleted for AMPK, we showed that metformin treatment induced lipid accumulation and autophagy inhibition independently of AMPK expression. (Attached Figure 4a-c). At the discretion of the reviewer, this figure can be added as supplemental figure. Finally, metformin was described to inhibit mTORC pathway (*Kalender A et al, Cell Metabolism, 2010*)²⁹, and cannot be therefore responsible for autophagy inhibition since mTORC inhibition led to autophagy activation as said by reviewer#4.

References

1. Hamasaki, M. *et al.* Autophagosomes form at ER-mitochondria contact sites. *Nature* **495**, 389–393 (2013).
2. Garofalo, T. *et al.* Evidence for the involvement of lipid rafts localized at the ER-mitochondria associated membranes in autophagosome formation. *Autophagy* **12**, 917–935 (2016).
3. Hailey, D. W. *et al.* Mitochondria supply membranes for autophagosome biogenesis during starvation. *Cell* **141**, 656–667 (2010).
4. Wu, W. *et al.* FUNDC1 regulates mitochondrial dynamics at the ER-mitochondrial contact site under hypoxic conditions. *EMBO J.* **35**, 1368–1384 (2016).
5. Gomez-Suaga, P. *et al.* The ER-Mitochondria Tethering Complex VAPB-PTPIP51 Regulates Autophagy. *Curr. Biol. CB* **27**, 371–385 (2017).
6. Wieckowski, M. R., Giorgi, C., Lebiedzinska, M., Duszynski, J. & Pinton, P. Isolation of mitochondria-associated membranes and mitochondria from animal tissues and cells. *Nat. Protoc.* **4**, 1582–1590 (2009).
7. Scotland, S. *et al.* Mitochondrial energetic and AKT status mediate metabolic effects and apoptosis of metformin in human leukemic cells. *Leukemia* **27**, 2129–2138 (2013).
8. Farge, T. *et al.* Chemotherapy-Resistant Human Acute Myeloid Leukemia Cells Are Not Enriched for Leukemic Stem Cells but Require Oxidative Metabolism. *Cancer Discov.* **7**, 716–735 (2017).
9. Bosc, C., Selak, M. A. & Sarry, J.-E. Resistance Is Futile: Targeting Mitochondrial Energetics and Metabolism to Overcome Drug Resistance in Cancer Treatment. *Cell Metab.* **26**, 705–707 (2017).
10. Viale, A. *et al.* Oncogene ablation-resistant pancreatic cancer cells depend on mitochondrial function. *Nature* **514**, 628–632 (2014).
11. Carracedo, A., Cantley, L. C. & Pandolfi, P. P. Cancer metabolism: fatty acid oxidation in the limelight. *Nat. Rev. Cancer* **13**, 227–232 (2013).
12. Sawyer, B. T. *et al.* Targeting fatty acid oxidation to promote anoikis and inhibit ovarian cancer progression. *Mol. Cancer Res. MCR* (2020) doi:10.1158/1541-7786.MCR-19-1057.
13. Pascual, G. *et al.* Targeting metastasis-initiating cells through the fatty acid receptor CD36. *Nature* **541**, 41–45 (2017).
14. Lee, C.-K. *et al.* Tumor metastasis to lymph nodes requires YAP-dependent metabolic adaptation. *Science* **363**, 644–649 (2019).
15. Samudio, I. *et al.* Pharmacologic inhibition of fatty acid oxidation sensitizes human leukemia cells to apoptosis induction. *J. Clin. Invest.* **120**, 142–156 (2010).
16. Sui, X. *et al.* Metformin: A Novel but Controversial Drug in Cancer Prevention and Treatment. *Mol. Pharm.* **12**, 3783–3791 (2015).
17. Viollet, B. *et al.* Targeting the AMPK pathway for the treatment of Type 2 diabetes. *Front. Biosci. Landmark Ed.* **14**, 3380–3400 (2009).
18. Csordás, G. *et al.* Structural and functional features and significance of the physical linkage between ER and mitochondria. *J. Cell Biol.* **174**, 915–921 (2006).
19. Basso, V. *et al.* Regulation of ER-mitochondria contacts by Parkin via Mfn2. *Pharmacol. Res.* **138**, 43–56 (2018).

20. Le Marchand-Brustel, Y., Gautier, N., Cormont, M. & Van Obberghen, E. Wortmannin inhibits the action of insulin but not that of okadaic acid in skeletal muscle: comparison with fat cells. *Endocrinology* **136**, 3564–3570 (1995).
21. Loubière, C. *et al.* Metformin-induced energy deficiency leads to the inhibition of lipogenesis in prostate cancer cells. *Oncotarget* **6**, 15652–15661 (2015).
22. Regazzetti, C. *et al.* Regulated in development and DNA damage responses -1 (REDD1) protein contributes to insulin signaling pathway in adipocytes. *PloS One* **7**, e52154 (2012).
23. Ouimet, M. *et al.* Autophagy regulates cholesterol efflux from macrophage foam cells via lysosomal acid lipase. *Cell Metab.* **13**, 655–667 (2011).
24. Singh, R. *et al.* Autophagy regulates lipid metabolism. *Nature* **458**, 1131–1135 (2009).
25. Molina, J. R. *et al.* An inhibitor of oxidative phosphorylation exploits cancer vulnerability. *Nat. Med.* **24**, 1036–1046 (2018).
26. Sriskanthadevan, S. *et al.* AML cells have low spare reserve capacity in their respiratory chain that renders them susceptible to oxidative metabolic stress. *Blood* **125**, 2120–2130 (2015).
27. Skrtić, M. *et al.* Inhibition of mitochondrial translation as a therapeutic strategy for human acute myeloid leukemia. *Cancer Cell* **20**, 674–688 (2011).
28. Lee, E. A. *et al.* Targeting Mitochondria with Avocatin B Induces Selective Leukemia Cell Death. *Cancer Res.* **75**, 2478–2488 (2015).
29. Kalender, A. *et al.* Metformin, independent of AMPK, inhibits mTORC1 in a rag GTPase-dependent manner. *Cell Metab.* **11**, 390–401 (2010).

Reviewers' Comments:

Reviewer #1:

Remarks to the Author:

The authors have done an excellent job in responding my concerns. Most notably they have addressed the concern on the causality of MERCs to regulate autophagy and lipid accumulation through up and down experimental modulation of ER-mitochondria interactions. I will appreciate whether the data on MERC disruption by IP3R1 silencing (an ER protein rather than a mitochondrial protein) could be added in the manuscript as supplemental figures to strengthen conclusions.

In addition, functional analysis of MERCs by measurement of mitochondrial calcium levels also provide supportive evidence for structural analysis, even it is a pity that the authors did not specifically access IP3R-mediated mitochondrial calcium accumulation. Basal mitochondrial levels are linked by different calcium inputs and outputs, and it does not only depend of ER-mitochondria calcium transfer. Therefore, it would be fairer to mention this limitation and to change the following sentence "This thus indicates a MERCs-dependent diminished Ca²⁺ transfer from ER to mitochondria" by "This thus suggests a MERCs-dependent diminished Ca²⁺ transfer from ER to mitochondria".

Lastly, the authors have made the effort to perform subcellular fractionation, which I agree is a challenging strategy in vitro! A surprising fact is the absence of VDAC and IP3R1 at MERC fractions, despite the presence of FALC4. I am afraid that this will surprise the scientific community working on MERCs, as the VDAC-Grp75-IP3R1 is the more admitted calcium channelling complex at MERCs. Maybe it is related to the quantity of protein loading and/or the time of exposition of membranes during western blotting. This point should be either discussed or addressed.

Reviewer #2:

Remarks to the Author:

In the revised manuscript, the authors have put in a considerable amount of effort to strengthen the story. Most of my concerns have been addressed in this revised manuscript. However, I was unable to access the Attached Figure 2 the authors refer to in the rebuttal. The figure is missing, and the reviewer is requesting access to this Figure panel.

Two minor points:

- Please refer to the LC3 flux as LC3 flux and not %.
- The graphical abstract is now substantially improved. My suggestion would be to incorporate autophagy and lysosomes on the right side in a graphical manner in addition to the label saying 'autophagy inhibition'.

Reviewer #3:

Remarks to the Author:

The authors have satisfied my concerns.

Reviewer #4:

Remarks to the Author:

Authors have performed additional experiments and my concerns have been addressed. I am satisfied for the revision.

REVIEWER COMMENTS

Reviewer #1 (Remarks to the Author):

The authors have done an excellent job in responding my concerns. Most notably they have addressed the concern on the causality of MERCs to regulate autophagy and lipid accumulation through up and down experimental modulation of ER-mitochondria interactions.

We would like to thank the reviewer once more for her/his constructive comments, which have greatly helped us to improve the manuscript. We have further addressed all remaining comments as below.

I will appreciate whether the data on MERC disruption by IP3R1 silencing (an ER protein rather than a mitochondrial protein) could be added in the manuscript as supplemental figures to strengthen conclusions.

As requested, the immunoblots shown in Attached Figure 1d in our first rebuttal letter have now been included in the manuscript (now Supplemental Figure 5f). These additional results are now commented in the results section (page 10).

In addition, functional analysis of MERCs by measurement of mitochondrial calcium levels also provide supportive evidence for structural analysis, even it is a pity that the authors did not specifically access IP3R-mediated mitochondrial calcium accumulation. Basal mitochondrial levels are linked by different calcium inputs and outputs, and it does not only depend of ER-mitochondria calcium transfer. Therefore, it would be fairer to mention this limitation and to change the following sentence “This thus indicates a MERCs-dependent diminished Ca²⁺ transfer from ER to mitochondria” by “This thus suggests a MERCs-dependent diminished Ca²⁺ transfer from ER to mitochondria”.

As suggested by the reviewer this sentence has been rephrased (see page 8 in the new version of the manuscript).

Lastly, the authors have made the effort to perform subcellular fractionation, which I agree is a challenging strategy in vitro! A surprising fact is the absence of VDAC and IP3R1 at MERC fractions, despite the presence of FALC4. I am afraid that this will surprise the scientific community working on MERCs, as the VDAC-Grp75-IP3R1 is the more admitted calcium channelling complex at MERCs. Maybe it is related to the quantity of protein loading and/or the time of exposition of membranes during western blotting. This point should be either discussed or addressed.

We do agree with the reviewer that the fact that VDAC1 and IP3R1 could not be detected at MERC fractions deserves clarification. We tried to overexpose the membranes and we could only detect a faint VDAC1 band in one of the experiments. Therefore, as mentioned by the reviewer, we do believe that the absence of detection of IP3R1 and VDAC1 is related to the low amount of protein retrieved at MERC fractions after fractionation. We are now commenting that point in the revised manuscript (page 8):

“To next validate the role of MERCs in autophagosome formation in AML cells, we developed a protocol based on Wieckowski *et al* subcellular fractionation procedure²⁹. We fractionated extracts of AML cells and characterized the MERCs fraction by the presence of FALC4 (Fig. 4c). Of note, the absence of detection of VDAC1 and IP3R1 in this fraction was likely due to the relative low amount of proteins obtained at the end of the procedure.”

Reviewer #2 (Remarks to the Author):

In the revised manuscript, the authors have put in a considerable amount of effort to strengthen the story. Most of my concerns have been addressed in this revised manuscript. However, I was unable to access the Attached Figure 2 the authors refer to in the rebuttal. The figure is missing, and the reviewer is requesting access to this Figure panel.

We would like to thank reviewer #2 for pointing out that our substantial revisions have addressed most of her/his concerns and thereby strengthened our conclusions.

Regarding the Attached Figure 2, we are sorry to read that it was not attached to the rebuttal letter. We do apologize for this inconvenience. The figure is now inserted below:

Attached Figure 2

Two minor points:

- Please refer to the LC3 flux as LC3 flux and not %.

As requested, we refer to the LC3 flux as LC3 flux in Figure 3c,d, in Supplemental Figure 3b and in Figure 5m.

- The graphical abstract is now substantially improved. My suggestion would be to incorporate autophagy and lysosomes on the right side in a graphical manner in addition to the label saying 'autophagy inhibition'.

We thank the reviewer for mentioning that our graphical abstract has been improved. As suggested, we are now including drawings of autophagosomes and lysosomes on the right side of the scheme in addition to the label saying 'autophagy inhibition'.

Reviewer #3 (Remarks to the Author):

The authors have satisfied my concerns.

We would like to thank reviewer#3 for pointing out that our revisions have thoroughly addressed her/his concerns.

Reviewer #4 (Remarks to the Author):

Authors have performed additional experiments and my concerns have been addressed. I am satisfied for the revision.

We thank reviewer#4 for stating that we have addressed her/his concerns.

Reviewers' Comments:

Reviewer #1:

Remarks to the Author:

The authors have satisfied my concerns.

Reviewer #2:

Remarks to the Author:

Contrary to what the authors reply, Figures 3c,e,f and Supplementary Figures 3b do not have LC3 flux calculations. This is a very important point because the authors have done the LC3 flux assay correctly. It is a matter of representing the results correctly too. To clarify again, simply changing the Y-axis label will be wrong because doing so will wrongly represent the 'LC3 flux' as <1 (that will mean there is no flux) while the data points to the opposite fact.

Point by point response to the referee comments

Reviewer #1 (Remarks to the Author):

The authors have satisfied my concerns.

We thank reviewer#1 for stating that we have addressed her/his concerns.

Reviewer #2 (Remarks to the Author):

Contrary to what the authors reply, Figures 3c,e,f and Supplementary Figures 3b do not have LC3 flux calculations. This is a very important point because the authors have done the LC3 flux assay correctly. It is a matter of representing the results correctly too. To clarify again, simply changing the Y-axis label will be wrong because doing so will wrongly represent the 'LC3 flux' as <1 (that will mean there is no flux) while the data points to the opposite fact.

We would like to thank the reviewer for her/his constructive comments that greatly helped us to improve the quality of the manuscript. We believe that we have now addressed her/his remaining concern. As requested, the autophagic flux now represents for each culture condition (e.g. +/- Met) the difference in the normalized amount of LC3II between chloroquine-treated and untreated cells. This calculation method does not change our conclusions since we still observe that the autophagic flux is decreased upon metformin treatment in AML cells (see below).